



# The ERA5-Land Soil-Temperature Bias in Permafrost Regions

Bin Cao[1], Stephan Gruber[2], Donghai Zheng[1], and Xin Li[1,3]

[1]National Tibetan Plateau Data Center, Institute of Tibetan Plateau Research, Chinese Academy of Sciences, Beijing, China
[2]Department of Geography and Environmental Studies, Carleton University, Ottawa, Canada
[3]CAS Center for Excellence in Tibetan Plateau Earth Sciences, Chinese Academy of Sciences, Beijing, China

**Correspondence:** Xin Li (xinli@itpcas.ac.cn), Donghai Zheng (zhengd@itpcas.ac.cn)

**Abstract.** ERA5-Land (ERA5L) is a reanalysis product derived by running the land component of ERA5 at increased resolution. This study evaluates its soil temperature in permafrost regions based on observations and published permafrost products. Soil in ERA5L is predicted too warm in northern Canada and Alaska, but too cold in mid-low latitudes, leading to an average bias of -0.08 °C. The warm bias of ERA5L soil is stronger in winter than in other seasons. Diagnosed from its soil temperature,

ERA5L overestimates active-layer thickness and underestimates near-surface (< 1.89 m) permafrost area. This is, in part, due to the shallow soil column and coarse vertical discretization in the ERA5 land-surface model and to warmer simulated soil. The soil-temperature bias in permafrost regions correlates with the bias in air temperature and with maximum snow height. Review of the ERA5L snow scheme and a simulation example point to a low bias in ERA5L snow density as a possible cause for warm-biased soil. The apparent disagreement of station-based and spatial evaluation of ERA5L highlights challenges in our

ability to test permafrost simulation models. While global reanalyses are important drivers for permafrost simulation, ERA5L soil data is not well suited for directly informing permafrost research decision making. To alleviate this, future soil-temperature products in reanalyses would require permafrost-specific alterations to the land-surface models used.

## 1   Introduction

Permafrost regions occupy more than one fifth of the exposed land area in the Northern Hemisphere and are subject to important

temperature dependent processes (Cheng and Wu, 2007; Westermann et al., 2009; Schuur et al., 2015; Walvoord and Kurylyk, 2016). As permafrost research is often impeded by sparse observations, global simulation products can be an important source of insight if their suitability can be established. Correspondingly, this study investigates the accuracy of soil temperature from the new ERA5-Land (ERA5L) high resolution reanalysis with a focus on permafrost area.

Reanalysis, assimilating a broad range of observations into fully coupled process-based models (land-atmosphere-ocean-sea

ice, and often biogeochemical components), is a valuable source of data for permafrost science. It has been successfully used in analyzing and simulating various permafrost phenomena, such as spatial distribution (e.g., Cao et al., 2019b; Fiddes et al., 2015; Slater and Lawrence, 2013), thermal state (e.g., Guo and Wang, 2017; Koven et al., 2013), active layer thickness (e.g., Tao et al., 2018; Qin et al., 2017), ground ice loss (e.g., Aas et al., 2019), and carbon release (e.g., Koven et al., 2015) at different scales. However, such applications are mostly restricted to using atmospheric variables as model forcing. By contrast,

soil temperature in reanalysis is rarely used directly due to the coarse spatial resolution (50–150 km) and bias. For example,



over the Qinghai–Tibetan Plateau (QTP), Hu et al. (2019); Yang and Zhang (2018) reported that the root mean squared error (RMSE) of daily soil temperature was up to 1.8–5.1°C, and generally the soil temperature from different reanalyses (i.e. ERA5-Interim/Land, MERRA-2, and CFSR) is consistently cold bias.

ERA5 is the latest reanalysis of the European Centre for Medium-Range Weather Forecasts (ECMWF). Compared to ERA-Interim, it includes new observations and revised processes, such as surface runoff and snow thermal insulation (ECMWF, 2018). Cao et al. (2019a) proposed the suitability of ERA5 data as forcing for permafrost temperature simulation, and its improved performance of atmospheric component in high latitudes has been reported (Graham et al., 2019). More recently, ERA5L was released as an improved land component of ERA5. Particularly, the new soil and snow hydrology (Balsamo et al., 2009; Dutra et al., 2010), revised soil thermal conductivity (Peters-Lidard et al., 1998), vegetation seasonality (Boussetta et al., 2013), and bare soil evaporation (Albergel et al., 2012) likely make it more accurate for many land applications. With a spatial resolution of 0.1°, ERA5L is the first global reanalysis product at an intermediate spatial scale between Earth-system land-surface models (e.g., Melton et al., 2019; Chadburn et al., 2015) and statistical and/or remote sensing-based permafrost products (e.g., Obu et al., 2019; Karjalainen et al., 2019b).

Here, we evaluate the soil temperature of ERA5L in permafrost regions against observations and against other published permafrost products. Furthermore, we investigate temperature bias using statistical analysis and example simulations at a well-instrumented location. The objectives of this study are to (1) assess the accuracy of ERA5L soil temperature in permafrost regions and (2) discuss the usability of ERA5L for permafrost research in light of the revealed bias and its potential causes.

## 2  Data

### 2.1  ERA5 and ERA5-Land

ERA5 is the latest generation atmospheric reanalysis produced by ECMWF. Data currently covers from 1979 onward and is expected to be available starting in 1950. ERA5 is produced using 4D-Var data assimilation in ECMWF's Integrated Forecast System, with a horizontal resolution of 0.25° (31 km), a temporal resolution of 1 hour, and a vertical resolution of 137 hybrid sigma model levels. The 37 pressure levels of ERA5 are identical to ERA-Interim (Noël et al., 2019). ERA5 assimilates improved input data that better reflects observed changes in climate forcing, as well as many new or reprocessed observations that were not available during the production of ERA-Interim. Different from other reanalyses, ERA5 additionally provides an estimate of uncertainty based on a ten-member ensemble with a reduced temporal resolution of 3 hours and spatial resolution of 0.5°(Albergel et al., 2018).

ERA5L is based on running the land component of the model driven by, but without coupling to, the atmospheric models. It uses the Tiled ECMWF Scheme for Surface Exchanges over Land with a revised land-surface hydrology (HTESSEL, CY45R1). ERA5L is forced by the atmospheric analysis of ERA5 and hence observations indirectly influence the simulations. It is delivered at the same temporal resolution as ERA5 and with a higher spatial resolution of 0.1°. ERA5L is currently available for 2001–2018, and it will eventually be extended back to 1950 and updated to the present time with little delay.



## 2.2 HTESSEL

### 2.2.1 Snow scheme

Compared to ERA-Interim, a more realistic representation of snow is used in HTESSEL. It is treated as a single layer above soil with independent prognostic temperature, mass, density, and albedo (Orsolini et al., 2019). The description of snow processes in HTESSEL by Dutra et al. (2010) can be summarized as: (1) liquid water with phase changes coexists with ice in the snow pack and is diagnosed from its temperature, mass, and density (B1); (2) density changes with overburden, thermal metamorphsim and retained liquid water following Lynch-Stieglitz (1994) (B1); (3) albedo changes exponentially with snow age and is adjusted

by vegetation condition; (4) snow cover fraction depends on both snow water equivalent (SWE) and density (B2).

### 2.2.2 Soil scheme

The soil heat transfer of ERA5L is governed by the Fourier law. While the thermal effects associated with latent heat are accounted for following Rouse (1984), soil thermal conductivity depends on moisture content only, and the influence of phase change is not represented. The upper boundary is a heat flux at the ground surface, derived from a weighted average over eight

subgrid fractions (or "tiles"). Zero heat flow is assumed at the lower boundary. The soil column of ERA5L is discretized into four layers with node depths (layer boundaries) of 0.07 (0–0.07), 0.21 (0.07–0.28), 0.72 (0.28–1.00), and 1.89 (1.00–2.89) m.

### 2.3 Observations and quality control

Soil temperatures from 639 stations located in permafrost regions are used (Table 1, Figure A1. See station metadata from supplement). These are 56 stations from the China Meteorological Administration (CMA, Wang et al., 2015), 105 stations from

World Data Centers (WDC) in Russia and Ukraine, 219 stations from Nordicana D, 95 stations from the Geophysical Institute, University of Alaska Fairbanks (GI-UAF), 10 stations from Tibetan Plateau observatory of plateau scale soil moisture and soil temperature (Tibet-Obs) (Su et al., 2011), 60 stations from multiscale Soil Moisture and Temperature Monitoring Network in the Central Tibetan Plateau (CTP-SMTMN) (Yang et al., 2013), 40 stations from the Global Terrestrial Network for Permafrost (GTN-P, Biskaborn et al., 2015), 28 stations from National Park Services (NPS) in Alaska (Wang et al., 2018), 16 stations from

the U.S. Geological Survey (USGS, Urban and Clow, 2017; Wang et al., 2018), 8 stations from HiWATER (Che et al., 2019), and 2 stations from Boike et al. (2018, 2019). The site permafrost zone information is derived from the digitized Circum-Arctic Map of Permafrost and Ground-Ice Conditions (denoted as IPA map, Zhang et al., 2000). The observed mean daily soil temperature of these stations ranges from -42 to 38 °C with the elevation range of about 0–5500 m. Additional evaluation with 931 stations in non-permafrost regions is conducted for comparison. All the temperature time series are visually checked in

order to remove obvious outliers. Observed active-layer thickness (ALT) from Peng et al. (2018) are used.





**Table 1.** Summary of soil temperature observations, including: station number (N) in permafrost regions, temporal and temperature coverage, and covered ERA5L soil layer (SL).)

| Source | N | Coverage | SL | Reference |
|---|---|---|---|---|
| CMA | 56 | 2001–2006 (-26–38) | 1–4 | Wang et al. 2015 |
| WDC | 105 | 2001–2015 (-40–30) | 2–4 | – |
| Nordicana D | 219 | 2001–2018 (-42–25) | 1–4 | – |
| GI-UAF | 95 | 2001–2018 (-40–23) | 1–4 | Wang et al. 2018 |
| Tibet-OBS | 10 | 2008–2016 (-18–28) | 1–3 | Su et al. 2011 |
| CTP-SMTMN | 60 | 2010–2016 (-15–20) | 1–3 | Yang et al. 2013 |
| GTN-P | 40 | 2001–2018 (-41–26) | 1–4 | Biskaborn et al. 2015 |
| NPS | 28 | 2004–2016 (-33–24) | 2–3 | Wang et al. 2018 |
| USGS | 16 | 2001–2015 (-31–25) | 1–2 | Urban and Clow2017 |
| HiWATER | 8 | 2012–2017 (-19–22) | 1–3 | Che et al. 2019 |
| Others | 2 | 2001–2018 (-32–14) | 1–4 | Boike et al. 2018; 2019 |

## 2.4 Existing permafrost maps

Four permafrost maps are used here to compare permafrost area diagnosed from ERA5L soil temperature. They are (1) the IPA map compiled based on observations and mean annual air temperature (MAAT); (2) the heuristic 1-km global zonation index map from Gruber (2012) (denoted as PZI map), (3) the 1-km Northern Hemisphere permafrost map (Obu et al., 2019)

based on the semi-physical Temperature at the Top Of the Permafrost table (TTOP) model (TTOP map) driven by Moderate Resolution Imaging Spectroradiometers (MODIS) land surface temperature, and (4) the 1-km circumpolar permafrost map (CP map) derived from a statistical model (Karjalainen et al., 2019a). While ERA5L, TTOP and CP maps represent permafrost distribution with binary information (i.e. presence or absence based on soil temperature), the IPA map and PZI map use categories (e.g., continuous, discontinuous, sporadic, and isolated permafrost) or a continuous index (0.01–1) to represent the

proportion of an area underlain by permafrost (i.e permafrost extent). By following Melton et al. (2019), a threshold of 50% (continuous and discontinuous permafrost zones) for the IPA map and 0.5 for the PZI map, respectively, are used for permafrost area estimation and for comparing areas with binary maps.

## 3 Method

### 3.1 Evaluation

The observed temperatures are grouped by depth according to the four ERA5L soil layers. For the layer with observations from multiple depths, the one nearest to the ERA5L grid center is selected. ERA5L soil temperature is nearest-neighbour interpolated to observed sites to avoid the missing values of adjacent water body. The mean bias (BIAS), mean absolute error





(MAE), and RMSE were used for comparison against observations at station scale (see appendix A). As multiple sites could be located in the same grid cell, BIAS, MAE, and RMSE are calculated for each site and then aggregated for each unique grid by averaging with equal weights in each grid cell. In this context, weighted metrics, for example wBIAS, is used for the evaluation at ERA5L grid scale. MAAT bias and maximum snow depth ($SD_{max}$) were selected as candidate variables to be assessed as possible predictors of ERA5L temperature bias. $SD_{max}$ is derived as the median of annual maximum monthly snow depth during 2001–2018. Surface offset (SO) quantifies the influence of surface conditions, e.g., snow and vegetation cover (Smith and Riseborough, 2002), and is derived here as the difference of MAAT and mean annual ground temperature (MAGT) of soil layer 1 in ERA5L.

ERA5L permafrost is limited to the "near-surface" due to the shallow simulation depth, hence only sites with shallow ALT (< 1.89 m) are evaluated here. Near-surface permafrost is diagnosed from ERA5L soil temperature in two ways: (1) if soil at a depth of the four soil layers has an hourly temperature below 0 °C for two consecutive years ($ERA5L_H$); (2) if MAGT of the fourth layer is below 0 °C for two consecutive years ($ERA5L_A$), it is considered as permafrost.

### 3.2 Detailed permafrost simulation example

The suitability of ERA5L soil temperature and the effect of snow-density bias are further investigated with a site specific simulation example at a densely instrumented location near Lac de Gras (LdG), N.W.T., Canada (Figure 1A). GEOtop 2.0 (Endrizzi et al., 2014), a process-based numerical model, is used to simulate snow characteristics and soil temperature for ten terrain types from September 2015 to August 2017 as described in more detail by Cao et al. (2019a). The snow-correction factor (SCF) is used to scale modeled snow mass via precipitation. It is used as a lumped variable for representing precipitation bias in the driving reanalysis as well as differences between terrain types that are caused by preferential deposition and lateral transport by snow drifting. The ERA5 reanalysis and its ten-member ensemble are used as forcing.

## 4 Results

### 4.1 Soil temperature

ERA5L MAGT in the four soil layers has an overall wMAE of 2.52 °C and a wRMSE of 2.60 °C (Table 2). Soil temperature is found too high in western Canada and Alaska but too cold in mid-low latitudes, such as the the QTP, leading to a near-zero wBIAS of -0.08 °C (Figure 3). Among the 932 MAGTs from 331 ERA5L grid cells, 20.7% have RMSE less than 1 °C, 53.5% are better than 2 °C, and 68.9% are better than 3 °C.

The linear model is used here to predict ERA5L soil temperature bias. It is fitted using the 239 grid cells with both MAAT and MAGT leading to the following results:

$$wBIAS = 0.76 wBIAS_{MAAT} + 0.77 wSD_{max} + 0.15 \qquad (1)$$

where $wBIAS_{MAAT}$ is the weighted bias of MAAT. This model has an $R^2$ of 0.47 with $p < 0.01$ for both predictors. The result indicates both MAAT and snow depth have important influences on ERA5L soil temperature: (1) an increase of 1 °C in



**Table 2.** Comparisons of ERA5L mean annual air temperature (MAAT), mean annual ground temperature (MAGT) for different depths, and surface offset (SO) against observations and published data products.

| Metrics | | Permafrost region | | | | Non-permafrost region | | | |
|---|---|---|---|---|---|---|---|---|---|
| | | wBIAS | wMAE | wRMSE | N (site, grid) | wBIAS | wMAE | wRMSE | N (site, grid) |
| MAAT | | -1.05 | 1.88 | 1.93 | 2208 (268, 242) | -0.65 | 1.21 | 1.24 | 6095 (829, 828) |
| SO | | 0.41 | 1.84 | 1.94 | 268 (78, 67) | -0.83 | 1.10 | 1.14 | 2662 (584, 583) |
| MAGT | SL1 | -0.67 | 3.12 | 3.17 | 1144 (262, 173) | -1.74 | 2.04 | 2.07 | 2761 (627, 611) |
| | SL2 | 0.03 | 2.49 | 2.57 | 2330 (472, 283) | -1.43 | 1.73 | 1.78 | 5259 (833, 824) |
| | SL3 | -0.32 | 2.28 | 2.36 | 2070 (338, 261) | -1.51 | 1.77 | 1.83 | 4899 (791, 782) |
| | SL4 | -0.67 | 2.38 | 2.47 | 1658 (248, 215) | -1.69 | 1.92 | 1.98 | 4642 (763, 763) |
| | Overall | -0.08 | 2.52 | 2.60 | 7202 (556, 331) | -1.52 | 1.83 | 1.88 | 17561 (867, 850) |
| $MAGT_{avg}$ | ERA5L | -0.49 | 2.15 | 2.93 | 1626 (242, 209) | -1.47 | 1.68 | 2.38 | 3901 (581, 581) |
| | CP | -1.29 | 1.84 | 2.62 | | -1.55 | 1.71 | 2.32 | |
| | TTOP | -1.91 | 2.42 | 3.30 | | -0.38 | 1.28 | 1.94 | |

N is the total number of observations, annual or as averages over many years, with the number of sites and unique grid cells in parentheses. SL with subscripts 1–4 represent the soil layers of ERA5L, while "Overall" represent an average over the entire soil column. The $MAGT_{avg}$ is the average MAGT: 2001–2018 for ERA5L, 2000–2014 for the CP map, and 2002–2016 for the TTOP map.

**Table 3.** Comparisons of ERA5L permafrost area (PA) against previous estimates.

| Map | PA [$10^6$ km$^2$] | Diagnostic method | Period represented |
|---|---|---|---|
| ERA5L$_H$ | 5.5–7.6 | Subsurface hourly soil temperature $\leqslant$ 0 °C for two consecutive years | 2002–2018 |
| ERA5L$_A$ | 8.8–10.7 | Subsurface MAGT $\leqslant$ 0 °C for two consecutive years | 2002–2018 |
| TTOP | 13.9 | Equilibrium state model with MAGT < 0 °C | 2000–2016 |
| CP | 13.0–17.2 | Statistical model with MAGT < 0 °C | 2000–2014 |
| PZI | 12.9–17.7 | Heuristic-empirical model with PZI > 0.5 | a few decades prior to 1990 |
| IPA | 11.8–14.6 | Continuous and discontinuous permafrost zones | a few decades prior to 1990 |

Note that the CP map only represents permafrost distribution north of 30°N, TTOP map represents permafrost distribution of the Northern Hemisphere, and the others represent area of north of 60°S. Permafrost area from literature is given with their definition in this study.

MAAT wBIAS corresponds to an increase of 0.76 °C in ERA5L MAGT wBIAS; and (2) an increase of 1 m in snow depth
is equal to an increase of 0.77 °C in wBIAS. The overall wRMSE of SO is 1.94 °C and wBIAS is 0.21 °C which is found comparable to that of the land surface scheme (JULES) of UK Earth system model (UKESM) (Chadburn et al., 2015).

Averaged MAGTs from the CP and TTOP map are bilinerly interpolated to the observed sites and compared against the observations of the deepest soil layer. Note that the performance of CP and TTOP maps may be lower here than reported in

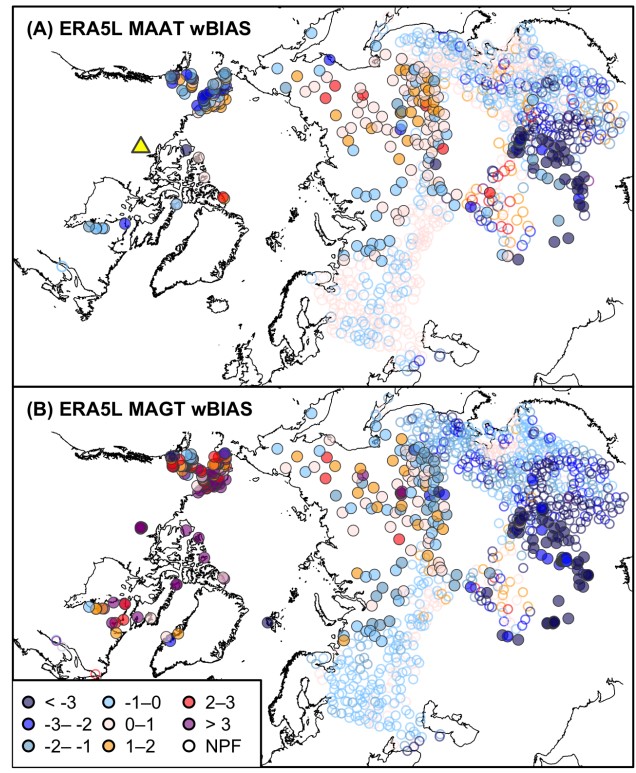

**Figure 1.** Comparison of ERA5L MAAT (A) and MAGT (B) against observations. wBIAS is simulated using all available MAGTs from four soil layers. Filled circle represents permafrost grids, while unfilled is the non-permafrost (NPF) one. Lac de Gras is marked as triangle in (A).

the original publications due to differing observations (depths, periods and proportion of sites in mountains) used. It is found
that ERA5L has an intermediate performance compared to them (Table 2). While Karjalainen et al. (2019b) found similar performance of their statistical model in predicting MAGT in permafrost and non-permafrost regions, our results show ERA5L and TTOP soil temperature have less agreement with observations in permafrost regions (Table 2, Figure 3). In addition to the worse performance of MAAT in these regions, the result suggests that more prevalent snow and soil freezing may reduce the suitability of HTESSEL for soil temperature simulation. The large warm bias of ERA5L soil temperature during winter
(Figure 2) further supports this notion.

## 4.2 Active-layer thickness and permafrost distribution

While ERA5L does not have the representation of deep ALT, our results show that even at the shallow ALT grids the mean ERA5L ALT (1.67 m) was more than 2 times of observed (0.82 m) (Figure 4). ERA5L ALT is substantially overestimated for most (72/79) of the grids, with wRMSE up to 0.98 m. Excluding glaciers, the mean near-surface permafrost area of the Northern
Hemisphere is estimated as $6.6\pm0.6\times10^6$ km$^2$ based on hourly soil temperature and $9.9\pm0.5\times10^6$ km$^2$ based on MAGT during



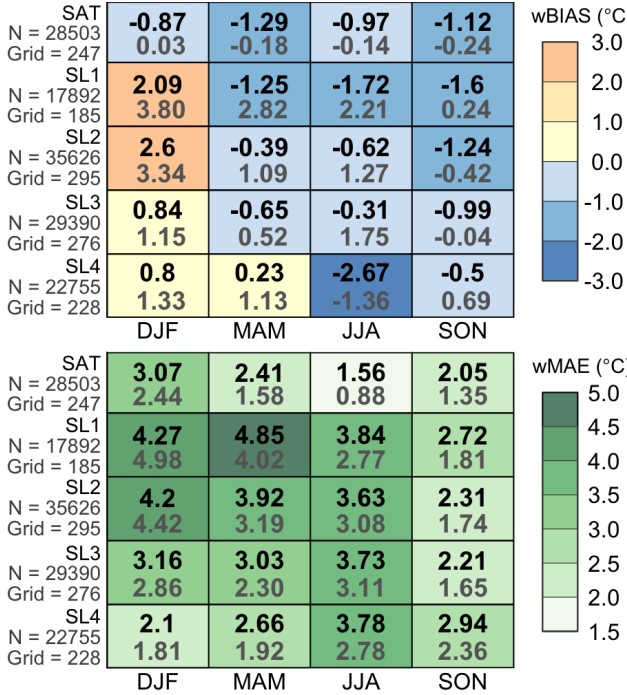

**Figure 2.** Monthly deviations of ERA5L soil temperature over permafrost regions. Monthly soil temperature is simulated for each depth and grid, and then the comparison is conducted for each season by averaging the MAE of all grids. The results of permafrost in Russia and Alaska is shown (in gray) in order to compare with Melton et al. (2019). SAT is the near-surface air temperature.

2002–2018 (Table 2, Figure 6). ERA5L underestimates permafrost area compared to earlier estimations (e.g., Zhang et al., 2000; Gruber, 2012; Obu et al., 2019; Karjalainen et al., 2019b). Near-surface permafrost area of ERA5L decreased with a rate of -0.11 (-0.08) $\times 10^6$ km$^2$ year$^{-1}$ based on hourly (annually) soil temperature (Figure 6). This corresponds to a loss of 1.7 (1.4) $\times 10^6$ km$^2$ of near-surface permafrost area since 2002.

### 4.3 Detailed permafrost simulation example

The detailed example simulation indicates that ERA5L soil temperature has warm bias (from 0.95 to 5.48 °C) in all terrain types, while GEOtop forced by ERA5 and its ten ensemble members show more reasonable results even when SCF = 1 (Figure 7). Specially, ERAL5 is found to only suitable in terrain types with exceptional snow deposition, e.g. in snowdrifts, tall shrubs, and sedge fen, and significantly warm-biased for the other terrain types during winter, and therefore, in the annual mean. While ERA5L SWE agrees with that of GEOtop when driven with the same data (SCF=1), its mean snow depth is approximate 1.53 times that of GEOtop and snow density is much lower (Table 4).



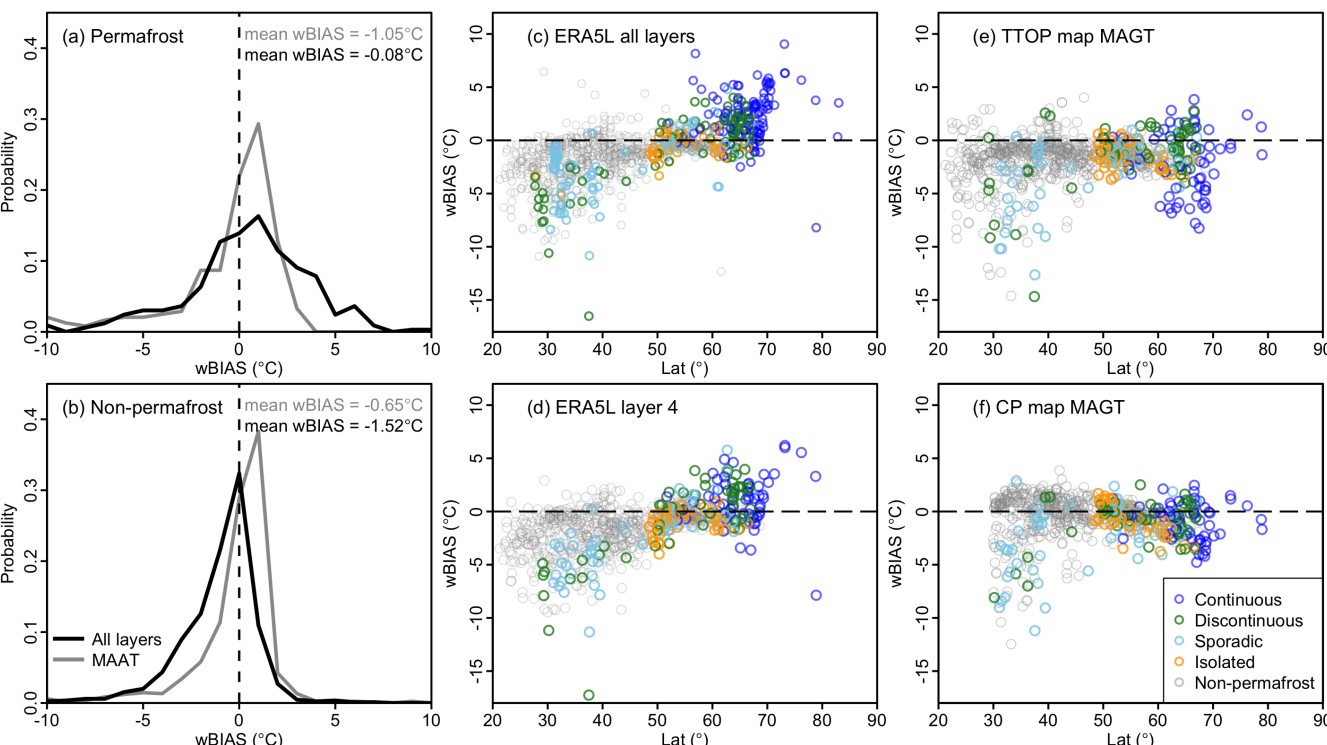

**Figure 3.** wBIAS (observation-ERAL) density of ERA5L mean annual air temperature (MAAT) and mean annual ground temperature (MAGT) in permafrost (a) and non-permafrost (b) regions as a whole. wBIAS of ERA5L overall MAGT (c), the last layer (d), TTOP map MAGT (e) and CP map MAGT (f) grouped by permafrost zone.

**Table 4.** Comparisons of September to March snow water equivalent (SWE, m), depth (m), and density (kg m$^{-3}$) near Lac de Gras in ERA5L and simulated with GEOtop driven by ERA5.

| Model | SWE | Depth | Density |
|---|---|---|---|
| ERA5L | 0.07 | 0.40 | 156 |
| GEOtop | 0.07 (0.01–0.1) | 0.27 (0.07–0.4) | 208 (160–226) |

The snow characteristics of GEOtop are derived using SCF = 1, and the range in
bracket reflects SCFs of 0.30–1.62, as used for the different terrain types in Figure 7.





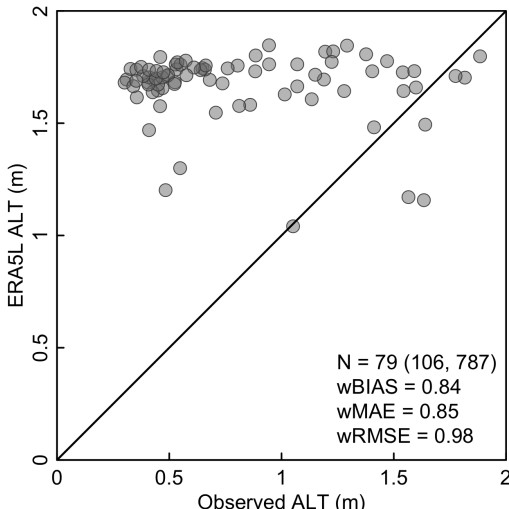

**Figure 4.** Comparison of shallow active layer thickness (ALT) based on 787 measurement from 106 stations located in 79 grids. The observed sites distribution is present in Figure 5.

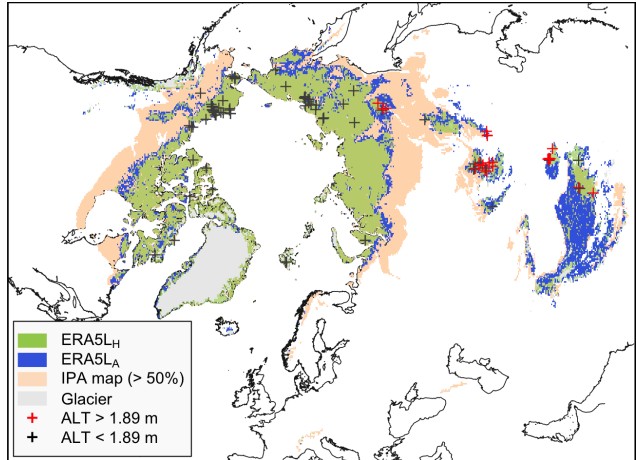

**Figure 5.** Near-surface permafrost area in 2002 derived from hourly (ERA5L$_H$) and annually (ERA5L$_A$) ERA5L soil temperature overlapping the continuous and discontinuous permafrost zones (permafrost extent > 50%) of IPA map. Active layer thickness (ALT) is from Peng et al. (2018).

## 5 Discussion

### 5.1 Suitability of ERA5L soil temperature

ERA5L has a number of advantages, such as long-term (back to 1950, eventually) and global coverage. While it could be seen
to provide an opportunity to study long-term changes of permafrost at an intermediate scale (∼9 km) without additional model simulation, our results indicate that significant bias in ERA5L soil temperature limits its utility for permafrost research.

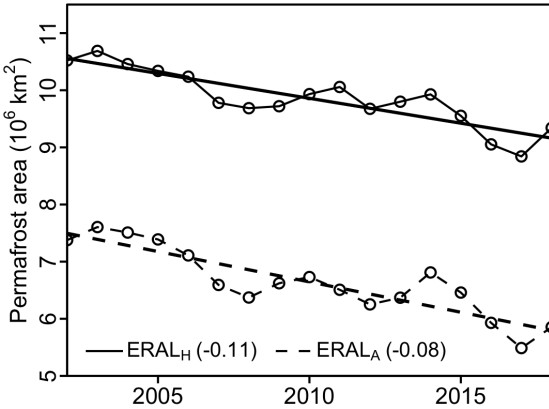

**Figure 6.** Changes of near-surface permafrost area between 2002–2018 derived from hourly (ERA5L$_H$) and annually (ERA5L$_A$) ERA5L soil temperature. Linear lines represent the trend of permafrost area based on linear model, and the rate is given in brackets.

Compared to the coarse-grid ($\sim$2.8°) simulation (Figure 4 from Melton et al., 2019), ERA5L often has more reasonable results in the deep soil layer although less permafrost processes are coupled, but worse in the shallow (Figure 2). ERA5L does not reproduce ALT well (Figure 4), likely due to its shallow soil column, coarse vertical discretization, warm bias in
soil temperature and lack of phase-dependent thermal conductivity in soil. Furthermore, ERA5L shows remarkable low bias in estimated permafrost area (Table3, Figure 5) when compared with previous estimates. The reason is that the large ALT (i.e. > 1.89 m) that frequently develops in mid-latitude mountains (e.g., Zhao et al., 2010; Cao et al., 2017), cannot be represented by the shallow soil column of ERA5L. While this could result in a low bias of permafrost area on the QTP, where observed ALT is generally large, a cold bias of soil temperature is found here at the same time. On the other hand, ERA5L underestimates
permafrost area in Canada and Alaska although the observed ALT there is mostly low. This is because ERA5L soil temperature in western Canada and Alaska appears too warm with a wBIAS of about +1.5 °C.

Loss of permafrost is to be expected with a warming atmosphere. While the loss of near-surface permafrost area derived from ERA5L is similar to that in previous land-surface model simulations (Lawrence et al., 2008; Slater and Lawrence, 2013), the absolute numbers and the rate of loss, however, have little value for further interpretation. This is because permafrost area
has pronounced bias to begin with and its temporal dynamics are known to be badly represented with a shallow soil column and are likely subject to inadequate representation of snow. Furthermore, because permafrost extent is a variable that cannot be observed, we fundamentally lack possibilities for proper validation (Gruber, 2012).

## 5.2 Model evaluation with sparse data

Using summary statistics from 242 sites in 209 grid cells alone would misleadingly show ERA5L to have comparably good skill
in representing the thermal state of permafrost, for example outperforming the TTOP map in all evaluation metrics (Table 2). Its simulated permafrost area, however, is visibly low when plotted geographically (Figure 5). Both findings can be reconciled because warm bias at high latitudes and cold bias in mid-latitudes cancel out each other based on the observations available

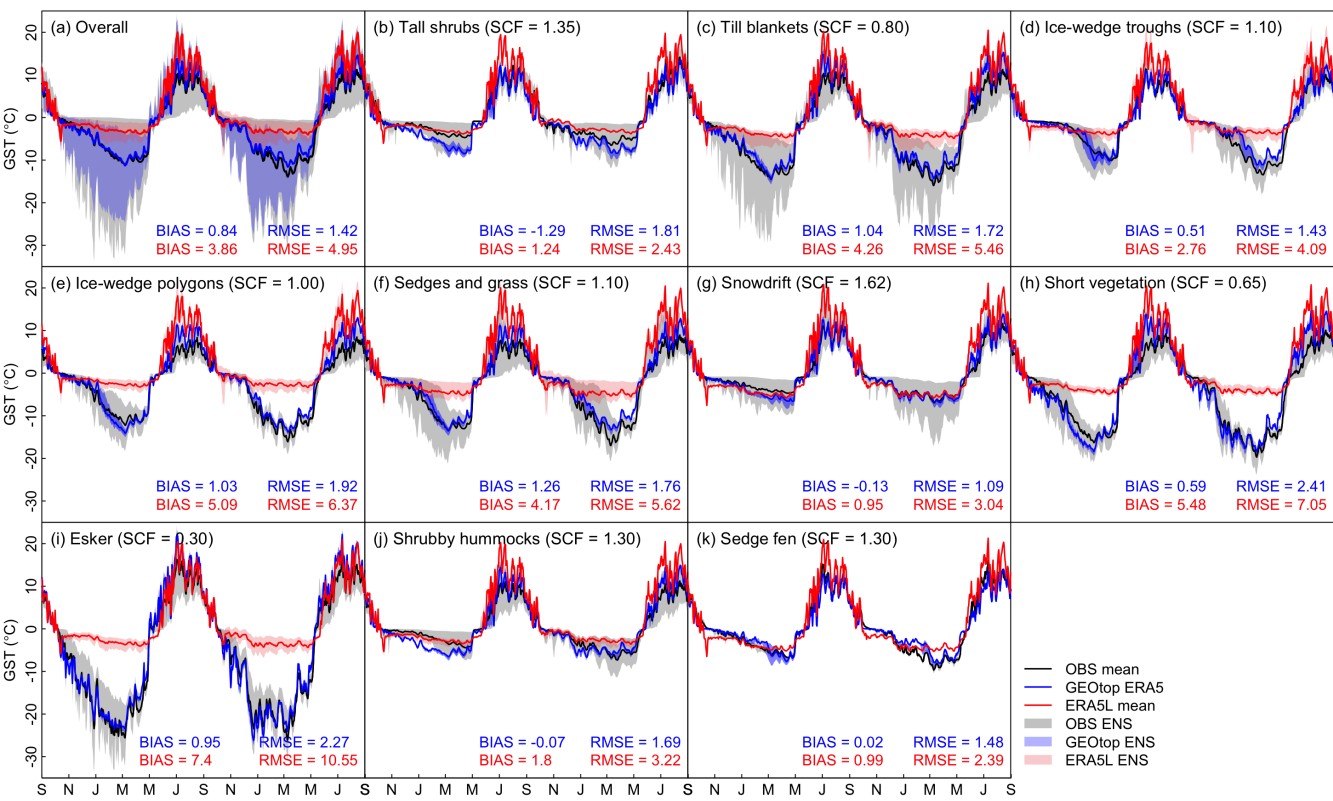

**Figure 7.** Ground surface temperature (GST) at the depth of 0.1 m for ten terrain types with different snow deposition in LdG, northern Canada. Observations and GEOtop ERA5 are from Cao et al. (2019a), while the GEOtop ENS is the ensemble range derived from ten-member ensemble of ERA5. Note that the first layer ERA5L soil temperature are used here. The BIAS and RMSE are simulated at a daily scale for each terrain type.

(Figure 3). Clearly, an improvement in summary statistics alone is not a sufficient criterion of superior model performance. Along these lines, the action group "Specification of a Permafrost Reference Product in Succession of the IPA Map" of the International Permafrost Association recommended in 2016 that in order to make progress, we needed the capability to measure whether a new map or model output was of superior quality compared with an old one and for this, the permafrost community needed to develop and provide the necessary data, methods, and standards (Gruber, 2016).

## 5.3 Scale effects

Even for an small area that is within a single grid cell of Earth-system models or reanalyses (10–100 km), evaluation with point observations are difficult. This could be demonstrated by our simulation example at LdG. Within an area of about 20 km × 30 km, MAGT and SO vary by almost 7 °C (Cao et al., 2019a) based on plot sizes on the order of 15 m × 15 m (Gruber et al., 2018). This is important in two ways. First, the results from statistical evaluations of a coarse-scale products such as





ERA5L significantly depend on the local selection of observation sites. This issue is know as the spatial effect when the lack of spatially-distributed measurements consistent with the size of model grid cells (i.e. 0.1° in ERA5L) is a potential source
of error for model evaluation (Gupta et al., 2006; Gubler et al., 2011). Second, ERA5L ground temperatures can at best only represent a small fraction of the area within each of its grid cells and, as a consequence, their value as a permafrost climate service for informing, e.g., local decision making for adaptation, is limited.

## 5.4 Snow densification and heat transfer

The seasonal ERA5L soil temperature deviance (Figure 2A) and linear model (Eq.1) show a remarkable bias toward high
soil temperature in winter, correlated with snow height. While we do not imply that GEOtop-based simulations are correct or representing metamorphism in Arctic snow accurately (see Domine et al., 2019), they demonstrate that simulations with snow cover of similar mass but different density are able to match ground-temperature observations far better than ERA5L. Since snow thermal conductivity is described as a exponential formulation of its density (Eq. B12), the low-biased snow density of HTESSEL would contribute to a much lower snow thermal conductivity. Furthermore, with the same SWE, low-biased snow
density means high-biased snow depth. In this context, the temperature gradient, and hence the heat flux though the snow pack are then smaller. Using the mean snow density in Table 4 as an example, a snow density of 75% would reduce ground heat-loss though the winter to about 44%. Even though this is one local case study at LdG, it sheds light on what may be causing the bias revealed for ERA5L soil temperature in cold regions more broadly. Interestingly, HTESSEL and GEOtop use the same exponential formulation of snow thermal metamorphism proposed by Anderson (1976) but different parameters. HTESSEL
uses $460$ ($m^3$ $kg^{-1}$) for $c_\xi$ (Dutra et al., 2010), $10^4$ times of that in GEOtop (Endrizzi et al., 2014) and Anderson (1976). As a consequence, with snow density greater than $100$ kg $m^{-3}$, its change rate ($s^{-1}$) related to thermal metamorphism remains near zero in HTESSEL. While this may explain, at least in part, the bias in ERA5L snow density and soil temperature, it is unknown whether the excessively high value for HTESSEL is merely an error in the publication cited or whether it reflects the value in the code. An additional contribution to higher snow densities in tundra environments may be the effect of blowing snow (cf,
Pomeroy et al., 1993).

## 6 Conclusion

Our results support five conclusions.

1  ERA5L soil temperature has a warm-bias at high-latitude and a cold bias in mid/low-latitude high-elevation areas. The soil-temperature bias in permafrost regions correlates with bias in air temperature and with maximum snow height.
225    Seasonally, soil temperatures in winter are more strongly warm biased than in other seasons. With more prevalent snow and ice, ERA5L soil temperature matches observations less well in permafrost-affected regions than in non-permafrost conditions.





2 Permafrost area is strongly underestimated when derived from ERA5L soil temperature and its temporal trend cannot be interpreted with confidence due to the bias in absolute area as well as model limitations.

230    3 Active-layer thickness is overestimated when derived from ERA5L soil temperature. This is due to the warm-bias in simulations as well as the shallow soil column and coarse vertical discretization used.

4 ERA5L snow density is hypothesized to have a low bias, at least in high-latitude areas, explaining part of the warm bias in soil temperature.

5 Summary statistics of comparing ERA5L with other spatial permafrost data based on their skill in reproducing observa-

tions do not agree with a geographic comparison of permafrost zones that are known to exist with some (albeit difficult to quantify) confidence. Whereas ERA5L performs well in the statistical evaluation, it severely underestimates permafrost, especially in Canada and Alaska. This highlights the remaining challenges in developing data and procedures for testing permafrost simulation models and products.

While global reanalyses provide urgently needed meteorological drivers for permafrost simulation, their soil data is not well

suited for directly informing permafrost research or local adaptation decisions. As such, simulations using permafrost-specific land-surface models driven by reanalyses (Cao et al., 2019a; Fiddes et al., 2015) will likely gain importance. Making future soil-temperature products like ERA5L directly usable will require significant permafrost-specific alterations to the land-surface models used.

*Code availability.* The Python script for downloading ERA5-Land is developed from API request provided by ECMWF Climate Data Store

(CDS) service and is available from the supplement.

*Data availability.* Soil temperature over China is not publicly available but could be requested from National Meteorological Information Center (http://data.cma.cn/). The other datasets are open access (last access: 5 November 2019). WDC dataset is available from http://www. wdcb.ru/, GTN-P dataset is available from https://gtnp.arcticportal.org/, USGS dataset is available from https://pubs.er.usgs.gov/publication/ ds1021, NPS is available from https://irma.nps.gov/DataStore/, HiWATER dataset is from the Cold and Arid Regions Science Data Center

at Lanzhou (https://doi.org/10.3972/hiwater.001.2019.db), Tibet-Obs and CTP-SMTMN is available from National Tibetan Plateau Data Center (https://data.tpdc.ac.cn/zh-hans/data/ef949bb0-26d4-4cb6-acc2-3385413b91ee/). The Nordcana D data is available from http://www. cen.ulaval.ca/nordicanad/en_index.aspx, GI-UAF is available from Permafrost Laboratory of University of Alaska (https://permafrost.gi. alaska.edu/content/data-and-maps, and the datasets from Julia Boike is available from https://doi.pangaea.de/10.1594/PANGAEA.880120 and https://doi.pangaea.de/10.1594/PANGAEA.905236. The PZI and TTOP maps are available from their publication, and the IPA map is

available from National Snow & Ice Data Center https://nsidc.org/data/GGD318/versions/2.





## Appendix A: Evaluation metrics

$$BIAS = \frac{1}{N} \sum_{i=1}^{N} (T_{obs} - T_{mod}) \tag{A1}$$

$$MAE = \frac{1}{N} \sum_{i=1}^{N} (|T_{obs} - T_{mod}|) \tag{A2}$$

$$RMSE = \sqrt{\frac{\sum_{i=1}^{N} (T_{obs} - T_{mod})^2}{N}} \tag{A3}$$

where $T_{obs}$ is observed soil temperature and $T_{mod}$ is the temperature from ERA5-Land soil temperature, GEOtop, or literature.

## Appendix B: Snow scheme of HTESSEL

### B1  Snow densification

Snow density $\rho_s$ (km m$^{-3}$) is constrained to be between 50–450 kg$^{-3}$. The compaction of snow density, or change rate ($s^{-1}$), is parametrized as

$$\frac{1}{\rho_s} \frac{\partial \rho_s}{\partial t} = \frac{W_S}{\eta} + \xi_s + \frac{\partial L_s}{\partial t} \frac{1}{SWE - L_s} \tag{B1}$$

where the first term represents overburden, second term is thermal metamorphism (Anderson, 1976; Boone and Etchevers, 2001), and the last term is the influences of snow liquid water ($L_s$, kg m$^{-2}$) following Lynch-Stieglitz (1994). $W_s$ (Pa) is the pressure of overlying snow mass or snow water equivalent (SWE, m) , and $\eta$ (Pa s$^{-1}$) is the viscosity coefficient of snow.

$$W_S = \frac{1}{2} \cdot SWE \cdot g \tag{B2}$$

where $g$ is the acceleration of gravity of 9.807 (ms$^{-2}$). Snow viscosity is described as a function of snow temperature ($T_s$, K) and density following Anderson (1976)

$$\eta = \eta_0 \cdot \exp(a_\eta \cdot T_D + b_\eta \cdot \rho_s) \tag{B3}$$

where $\eta_0 = 3.7 \times 10^7$ (Pa s), $a_\eta = 0.081$ (K$^{-1}$), $b_\eta = 0.018$ (m kg$^{-3}$). $T_D$ (K) is the depression temperature,

$$T_D = 273.16 - T_s \tag{B4}$$

The change rate of $\rho_s$ related to thermal metamorphism is parametrized as

$$\xi_s = a_\xi \cdot \exp\left(-b_\xi \cdot T_D - c_\xi \cdot \Delta\beta_s\right) \tag{B5}$$



where the $a_\xi$, $b_\xi$, $c_\xi$, and $\rho_\xi$ are constant values of $2.8 \times 10^{-6}$ (s$^{-1}$), 0.042 (–), 460 (m$^3$ kg$^{-1}$), and 150 (kg m$^{-3}$) derived or modified from Anderson (1976) and Jordan et al. (1999). $\Delta\beta_s$ (kg m$^{-3}$) is given as

$$\Delta\beta_s = \begin{cases} \rho_s - \rho_\xi, & \rho_s > \rho_\xi \\ 0, & elsewhere \end{cases} \tag{B6}$$

where $\rho_\xi$ (kg m$^{-3}$) is equal to 150 kg m$^{-3}$. L$_s$ is diagnosed from snow temperature, SWE, and snow density,

$$L_s = f(T_s) \cdot L_s^c \tag{B7}$$

where $f(T_s)$ is the snow temperature function and $L_s^c$ is the snow liquid water capacity (kg m$^{-2}$).

$$f(T_s) = \begin{cases} 0, & T_s < T_f - 2 \\ 1 + \sin\left\{\frac{\pi(T_s - T_f)}{4}\right\}, & T_s \geq T_f - 2 \end{cases} \tag{B8}$$

where $T_f$ is 273.16 (K), $L_s^c$ is parameterized as a function of SWE and $\beta_s$,

$$L_s^c = SWE \cdot [r_l^{min} + (r_l^{max} - r_l^{min}) \cdot C] \tag{B9}$$

where $r_l^{min}$ and $r_l^{max}$ are constant values of 0.03 and 0.1, and $C$ is given as

$$C = \begin{cases} 0, & \beta_s > \beta_s^l \\ \frac{\beta_s^l - \beta_s}{\beta_s^l}, & \beta_s \leq \beta_s^l \end{cases} \tag{B10}$$

where $\beta_s^l$ is 200 (kg m$^{-3}$).

**B2   Snow cover fraction**

Snow cover fraction (SCF) can be given as

$$SCF = \frac{1}{SD_{cr}} \frac{SWE}{\rho_s} \tag{B11}$$

where $SD_{cr}$, the minimum snow depth that ensures complete coverage of the grid box, is set as 0.1 m.

**B3   Snow thermal conductivity**

By following Douville et al. (1995), the snow thermal conductivity ($\lambda_s$) is treated as a function of snow density,

$$\lambda_s = \lambda_i \left(\frac{\rho_s}{\rho_i}\right)^{1.88} \tag{B12}$$

where $\lambda_i$ is ice thermal conductivity of 2.2 (W m$^{-1}$ K$^{-1}$) and $\rho_i$ is ice density 920 (kg m$^{-3}$).





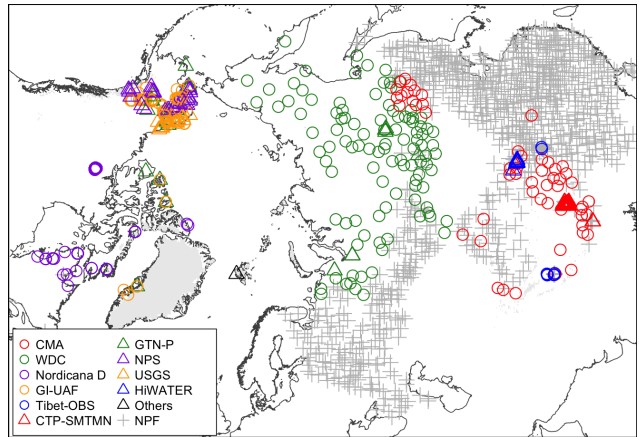

**Figure A1.** Distribution of soil temperature stations. Stations in permafrost regions are in color while the gray ones are non-permafrost (NPF) stations.

*Author contributions.* BC carried out this study by analyzing data, performing the simulations, organizing as well as writing the manuscript and was responsible for the compilation and quality control of the observations. SG proposed the initial idea, and contributed to organizing as well as writing the manuscript. XL and DHZ contributed to the writing the paper.

*Competing interests.* The authors declare that they have no competing interests.

*Disclaimer.* The authors declare that they have no conflict of interest.

*Acknowledgements.* The authors thank Joe Melton for his helpful comments. We thank Jaroslav Obu for the guideline of TTOP map, thank Kang Wang for introduction of soil temperature datasets over Alaska, thank Frank E. Urban and Gary D. Clow for providing access to the NPS datasets, thank Xiaoqing Peng for providing the active layer thickness datasets. ERA5-Land reanalysis data and ESA CCI LC map is provided by ECMWF. This study is funded by Strategic Priority Research Program of Chinese Academy of Sciences (grant no. XDA20100000, XDA20100104). Bin Cao is also supported by China Postdoctoral Science Foundation (grant no. 2019M660046).



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
