# Peer review of "The ERA5-Land Soil-Temperature Bias in Permafrost Regions"

_The Cryosphere, 2020_

## Referee Comment (RC1) · Anonymous Referee #1 · 14 May 2020

The study "The ERA5-Land Soil-Temperature Bias in Permafrost Regions" by Cao et al. evaluates the performance of the ERA5L reanalysis for ground temperatures and other ground-temperature-related parameters in permafrost areas. Although ground temperature is not a main target parameter for such reanalysis products, the study will be a valuable scientific contribution and I recommend publication after carefully revising the manuscript.

Major Comment/Recommendation:

When reading through the manuscript, many important points only became clear to me very late, i.e. in the Discussion. The temperature comparisons of the different products in the Results section, for example, left me wondering on the interpretation and implications. The same applies to the findings on the sizable reduction of "permafrost area" in

ERA5L, which only much later is resolved as likely being more an artefact of the model than reality. To a casual reader, the manuscript appears to make a number of potentially bold statements, without providing any hint that the interpretations/ clarification of implications are provided at some later stage in the Discussion (where some casual readers might miss it). While the strict separation of the different manuscript parts is in line with accepted methodology for scientific writing, I recommend guiding the reader through the manuscript in a better way. I have made more specific annotations and suggestions under general comments.

General comments:

Sect. 2.2 Remind the reader in one sentence what HTESSEL is, this is somewhat hidden in the previous text.

Sect. 2.3 and 3.1: Please add information on the depths of the available borehole temperatures and how this compares to the shallow ground representation in ERA5L. The Biskaborn-data set, for example, contains many borehole measurements at much deeper layers than ERA5L can represent, so (how) are these measurements used?

l.112: the first criterion is unclear, is this "if T of any of the four layers is constantly below zero for two years"?

Sect. 3.2 The added value of this is unclear at this stage of the manuscript, it seems to be rather unrelated to the main purpose, i.e. compare the direct ground T output of ERA5L to observations. This becomes clear only much later, but please add a few sentences on the purpose already here.

Table 2+3: add the references to the different products used (at least in the caption), so that the readers don't have to search for the abbreviations in the text.

l. 129: the purpose of the equation is unclear, and must be explained in more detail. If I understand correctly, you relate the bias in MAGT to the bias in MAAT, using the snow depth (which has no bias, I guess since measurements are not available?). Does

the intercept of 0.15 make sense, i.e. zero bias in MAAT and zero snow produces an MAGT bias of 0.15? Should one not rather prescribe an intercept of 0 in the equation? I guess it would not change much, considering the R2 of 0.47 of the relationship.

Table 2: I assume the comparison is done for individual years when- and wherever an entire year of observations is available? How does this relate to CP and TTOP which represent longer periods, are only observation that span the entire periods used? If not, to what extent does the availability of observations influence these comparisons - many observations are likely taken in recent years, which on average were warmer than earlier periods. There is the passage starting with "Note that the performance of CP and TTOP maps may be lower here than reported in...", but the implication of this is not really clear. Table 2 seems to suggest that ERA5L is considerably better than CP and TTOP for PF areas, but it is unclear if that conclusion can indeed be drawn. This is not only considering the study periods, but also the spatial distribution of the measurement sites (heavily biased towards China, SE Russia and Alaska according to Fig. 2). This point is adequately discussed in 5.2, but it would be good if some of it could be mentioned already here. At least include a statement "see Sect. 5.2 for a detailed discussion" in the text.

l. 137: typo "bilinearly" Fig. 1: add units in the figure.

Fig. 2 is only presented in one sentence in the text. This should be presented in more detail. I suggest using this to motivate Section 4.3 (see also comment above).

Table 4: Are there any snow density measurements from the site that could clarify which one of the models is right (or if both are wrong).

l. 152: Make it clear that this is "ERA5L PF extent as defined in this study", it is clear that the shallow soil column makes it very difficult to relate this to "true PF extent change". Such statements can easily be misunderstood, compare to "Lawrence, D.M. and Slater, A.G., 2005. A projection of severe near-surface permafrost degradation during the 21st century. Geophysical Research Letters, 32(24)." and the resulting comment by Burn &

Nelson. This issue is again explained later in the discussion, but make it clear already here, that this by no means represents real PF extent changes.

l. 168: what do you mean by "although less permafrost processes are coupled"?

L. 170: When I look at Fig. 5, I don't quite understand why there is a "remarkably low bias in PF extent". Your explanations later seem to go in the direction that this might rather be a coincidence, since biases in different regions cancel each other? Furthermore, ERA5L cannot really represent the discontinuous and continuous permafrost zones, so fractional PF coverage is by definition not included.

Sect. 5.4: Dedicated snow models like CROCUS and Snowpack also include formulations for compaction due to wind drift which likely occurs at LdG(?). If I understand correctly, this is neither included in the ERA5L model nor in GEOtop? This should be stated, especially since there seem to be no field measurements of snow densities from the site which could clarify which model is more right? I would certainly agree that the GEOtop snow densities look much more realistic, but that's more an educated opinion, rather than science.

Discussion general: Consider adding a Section "Implications" or similar – especially the findings on the snow cover and the shallowness of the ground representation are very interesting also for improvements of further reanalysis generations. To me it almost looks like that one might get a pretty good performance for permafrost parameters by doing a couple of obvious improvements of the ground and snow models (which likely wouldn't even cost a lot of additional computation). You study is a great reference for this, and stating this clearer will likely increase the impact of the paper.

---

## Referee Comment (RC2) · Anonymous Referee #2 · 5 Jun 2020

**General comments**

This paper presented a good assessment of the soil temperature at a large scale using in-situ observations and previous products/maps. Understanding current soil temperature bias in reanalysis could improve further Earth-system model design by accounting more essential permafrost processes and hence benefit the permafrost community. This paper is generally well written. I have some comments for further revisions.

**Major comment**

[Figure]

- As Reviewer#1 stated, some important points became clear a little bit late. To casual readers, this may be not easy to follow.

- The authors MUST recheck this statement in L70-71. From the ERA5L website, they said: "Temperature of the soil in layer 1 (0 - 7 cm) of the ECMWF Integrated Forecasting System. The surface is at 0 cm. Soil temperature is set at the middle of each layer, and heat transfer is calculated at the interfaces between them." This is very important because these depths were used to interpolate soil temperature profiles and to determine ALT, if my guess is correct. If incorrect depths were used, the comparisons were already artificially altered.

- The authors should describe the estimate of ALT by using ERA5L.

- Did the authors consider the uncertainties from vegetation?

- In section 2.3, I miss a description of air temperature observation, while it is used for analyses of ERA5L soil temperature bias (i.e. in Table 1 and the linear model). Authors have to add a brief description here, and even show them in a proper way. This could be easily done, for example, by changing the shape of the station with both air and soil temperatures in Figure A1.

**Specific comments:**

P2, L27: The RMSE of reanalyses soil temperature? Please clarify.

P2, L40: ... and example numerical or process-based simulation...

P2, L57: Note that ERA5L is now available from 1981.

P4, L86: The soil temperature from the TTOP and CP maps are used as comparisons, please as mention here.

P4, L89: ...(denoted as PZI map)",".., should it be ";"? Similar in L91.

P4, L97: The MAGT of TTOP and CP maps are additionally used as reference in your Table 1 and Figure 3. Please clarify here.

P5, L104: ...in the same ERA5L grid cell...

P5, L107: ...of ERA5L soil temperature....

P5, L126: there is a repeat of the "the".

P5, L134: ...and (2) an increase of 1 m wSDmax

P7, L149: Is the ALT also overestimated in high latitudes and underestimated in high altitudes?

P10, L164: Also mention the high spatial (and maybe temporal) resolution here, this is one of the most significant features of ERA5 compared to the others.

P13, L215: ...for $c_\xi$ in Eq. B5...

P13, L216: It should be 150 kg m$^{-3}$ based on Eq. B5, please double check.

P14, L236: Underestimate permafrost...(what)? Permafrost area? Please clarify.

P14, L252–253: The bracket is incomplete

P14, L255: Brackets are needed here for the url.

P15, L270: Add space between m and s$^{-2}$

P16, L278: $\rho_\xi$ is not included in Eq. (B5).

P16, L280: Considering move $\Delta\beta_s = 0$ to the upper so that Eq B6 would be aliened with the state of Eq. B8 and B10

P16, L297: ...ice density of 920...

**Tables and figures:**

- Table 1: This is only for the observations in permafrost regions. Please clarify in the caption otherwise including the observations in non-permafrost regions.

- Figure 3: In the caption, it should be "...(observation-ERA5L)..."

- Figure 6: Considering add unit to the permafrost area changing rate.

---

## Referee Comment (RC3) · Anonymous Referee #3 · 7 Jun 2020

This paper assesses the utility of ERA5L soil temperature products for permafrost studies by using a wide range of global station data from both permafrost and non-permafrost regions as well as detailed simulation experiments at a specific site. The authors find that ERA5L has large biases making the product problematic for permafrost studies. This study is a valuable contribution as we increasingly use reanalysis products for land surface modeling studies, especially at regional or global scales and insights into performance of these products are useful. Additionally, such studies may help to guide future developments in land surface schemes used in reanalyses. I recommend publishing after considering my (mainly minor) comments.

(in grammatical comments changes are CAPITALIZED)

1. l.3 "is predicted TO BE too warm...."

[Figure]

2. l.19 "Reanalysis, ASSIMILATES"

3. l.28 what is ERA5-Interim/Land? Seems a confusion of the products

4. l.29 "consistently cold BIASED."

5. l.54 I think the HTESSEL ref could do with a publication citation.

6. l.57 now available from 1981.

7. Section 2.2.1 what do B1 and B2 refer to?

8. l.71 is the node really at the lower boundary (0.07) in soil layer 1?

9. l.74 "These INCLUDE"

10. Section 2.3 and Table 1 are all stations boreholes? If so perhaps explicitly state that.

11. l.90-91 and driven by ERA-Interim air temperature.

12. l111-114: I don't quite understand the motivation for the two definitions of near-surface permafrost I think a sentence explaining why you do this would be helpful for the reader.

13. l129 "A linear model..."

14. l.137 What depth are these MAGT's? Averaged across time or space? Please provide a bit more detail here.

15. l.143 more prevalent snow and soil freezing in the model or in reality? Please clarify. If in reality, then permafrost regions do not necessarily have more prevalent snow than non-permafrost regions.

16. l.147 "While ERA5L does not have DATA allowing deep ALT values to be computed"

17. l.153 "(annually)" is it an annual average? Please clarify.

18 Figure 3 Interesting latitudinal trend in c,d. Can you shed more light on this in the discussion? I guess densification processes at high latitudes (badly represented wind?) What is driving the cold bias at low latitudes?

19. Figure 4 perhaps add the mean value that you cite in the text here.

20. l.170 "shows REMARKABLY"

21. l.194 "Even for A"

22. l.198 "This issue is KNOWN"

23. l.208 "as AN exponential..."

24. l.226 soil temperatureS MATCH..."

25. l.230 But what about the cold bias you see? the bias appears to evenly spread (figure3) why does this not give a similar spread in ALT estimates (Figure 4) and a related underestimation of ALT?

26, l.232. use of "low" here is confusing. you are biased to low densities, you do not have a low bias. I would say "a low-density bias" to make it clear.

---

## Author Comment (AC1) · 12 Jun 2020

**Responses of Anonymous Referee #1**

The authors would like to thank the reviewer for the constructive feedback, and the thorough assessment of the manuscript. Below we provide a point-to-point response to each comment, reviewer comments are given in black, responses are given in blue. Additionally, we have included details of how we intend to address these changes in a revised submission.

The study "The ERA5-Land Soil-Temperature Bias in Permafrost Regions" by Cao et

al. evaluates the performance of the ERA5L reanalysis for ground temperatures and other ground-temperature-related parameters in permafrost areas. Although ground temperature is not a main target parameter for such reanalysis products, the study will be a valuable scientific contribution and I recommend publication after carefully revising the manuscript.

**Major Comment/Recommendation:**

When reading through the manuscript, many important points only became clear to me very late, i.e. in the Discussion. The temperature comparisons of the different products in the Results section, for example, left me wondering on the interpretation and implications. The same applies to the findings on the sizable reduction of "permafrost area" in ERA5L, which only much later is resolved as likely being more an artefact of the model than reality. To a casual reader, the manuscript appears to make a number of potentially bold statements, without providing any hint that the interpretations/ clarification of implications are provided at some later stage in the Discussion (where some casual readers might miss it). While the strict separation of the different manuscript parts is in line with accepted methodology for scientific writing, I recommend guiding the reader through the manuscript in a better way. I have made more specific annotations and suggestions under general comments.

We agree, hints are added as suggested in the specific comments. Especially, Section 5.2 will be mentioned in the caption of Table 2 in order to avoid any possible misunderstanding. We'll also move the implication part from Section Conclusions to the Discussion.

**General comments:**
Sect. 2.2 Remind the reader in one sentence what HTESSEL is, this is somewhat hidden in the previous text.

In the revision, we'll change this part to "... a more realistic representation of snow is used in ERA5 land surface model of HTESSEL."

Sect. 2.3 and 3.1: Please add information on the depths of the available borehole temperatures and how this compares to the shallow ground representation in ERA5L. The Biskaborn-data set, for example, contains many borehole measurements at much deeper layers than ERA5L can represent, so (how) are these measurements used? Only the observed temperature within the ERA5L soil temperature column, i.e. 0–2.89 m, were used here. In Table 1, we'll add the depth range of used soil temperature observations for each data source. In Section 3.1, we'll add "Aligned ERA5L soil column, only the observed temperature within 0–2.89 m, were used here."

I.112: the first criterion is unclear, is this "if T of any of the four layers is constantly below zero for two years"?

Yes. It will be changed as "if soil at any depth of the four soil layers has an hourly temperature below  $0 \circ C$  for two consecutive years (ERA5LH);"

Sect. 3.2 The added value of this is unclear at this stage of the manuscript, it seems to be rather unrelated to the main purpose, i.e. compare the direct ground T output of ERA5L to observations. This becomes clear only much later, but please add a few sentences on the purpose already here.

At the beginning of Section 3.2, we'll add "Our results show remarkable bias of ERA5L soil temperature in winter that likely correlates with snow depth (Figure 2). For this reason, the suitability of ERA5L soil temperature and the effect of snow-density bias are further investigated with a site specific simulation example at a densely instrumented location near Lac de Gras (LdG), N.W.T., Canada (Figure 1A). This

TCD
detailed permafrost simulation example provides an opportunity to evaluate ERA5L soil temperature under different terrain (e.g. vegetation, soil properties) and snow conditions."

Table 2+3: add the references to the different products used (at least in the caption), so that the readers don't have to search for the abbreviations in the text. The reference will be added in the caption:

Table 2: "...The MAGTavg is the average MAGT: 2001–2018 for ERA5L, 2000–2014 for the CP map (Karjalainen et al., 2019), and 2002–2016 for the TTOP map (Obu et al., 2019)."

Table 3: "Note that the CP map only represents permafrost distribution north of 30° N (Karjalainen et al., 2019), TTOP map represents permafrost distribution of the Northern Hemisphere (Obu et al., 2019), and the others represent the area of north of 60° S. Permafrost area from literature is given with their definition in this study.

I. 129: the purpose of the equation is unclear, and must be explained in more detail. To clarify the purpose, we will

- 1) refer to Eq. 1 in Section 3.1: "MAAT bias and maximum snow depth (SDmax) were selected as candidate variables to be assessed as possible predictors of ERA5L soil temperature bias (see Eq. 1)".
- 2) this sentence is changed to "The linear model is used here to predict ERA5L soil temperature bias caused by MAAT bias and snow depth in permafrost regions.".

If I understand correctly, you relate the bias in MAGT to the bias in MAAT, using the snow depth (which has no bias, I guess since measurements are not available?).
**Yes, snow measurements are not available.**

Does the intercept of 0.15 make sense, i.e. zero bias in MAAT and zero snow produces an MAGT bias of 0.15? Should one not rather prescribe an intercept of 0 in the equation? I guess it would not change much, considering the R2 of 0.47 of the relationship.

We can expect uncertainty of the linear model with R2 of 0.47 since it was fitted with limited observations, i.e. 239 grid cells. However, the intercept of 0.15 makes sense. It means even no MAAT bias and snow cover is present, ERA5L soil temperature in permafrost regions could still have bias that may from the other variables, i.e. due to the mismatched depth of observations and ERA5L soil layer.

Table 2: I assume the comparison is done for individual years when- and wherever an entire year of observations is available?

Yes, for MAAT, SO, and MAGT evaluation, the comparison is done for individual available years, while the MAGTavg is the average MAGT for the entire long period. In the caption, we added "MAAT, SO, and MAGT were evaluated for the individual year, while  $MAGT_{avg}$  was carried through once for the entire period."

How does this relate to CP and TTOP which represent longer periods, are only observation that span the entire periods used? If not, to what extent does the availability of observations influence these comparisons - many observations are likely taken in recent years, which on average were warmer than earlier periods. There is the passage starting with "Note that the performance of CP and TTOP maps may be lower here than reported in: ::", but the implication of this is not really clear.

In Section 3.1, we'll add "The TTOP and CP map are derived using equilibrium model, and MAGT is given as an average of the entire period (MAGTavg), i.e. 2002–2014 for the CP map and 2002–2016 for the TTOP map, without uniform/specific soil depth. For better evaluation purpose, we aggregate all available observed MAGTs during the period by averaging, and then compared against the MAGTavg of these two maps. Note that the performance of CP and TTOP maps may be lower here than reported in
the original publications due to differing observations (depths, periods and proportion of sites in mountains) used." to clarify. The sentence of "Note that the performance of CP and TTOP maps may be lower here than reported in...", will be removed from Section 4.1

Table 2 seems to suggest that ERA5L is considerably better than CP and TTOP for PF areas, but it is unclear if that conclusion can indeed be drawn. This is not only considering the study periods, but also the spatial distribution of the measurement sites (heavily biased towards China, SE Russia and Alaska according to Fig. 2). This point is adequately discussed in 5.2, but it would be good if some of it could be mentioned already here. At least include a statement "see Sect. 5.2 for a detailed discussion" in the text.

Yes, the summary statistics with sparse data would be misleading. In the revision, we'll add "Note that the summary statistics present here are based on sparse data and need to be interpreted in light of the considerations outlined in Section 5.2." in the caption of Table 2.

I. 137: typo "bilinearly" Will be revised.

Fig. 1: add units in the figure. The unit will be included in the legend.

Fig. 2 is only presented in one sentence in the text. This should be presented in more detail. I suggest using this to motivate Section 4.3 (see also comment above). Figure 2 will be added in Section 3.2–"*Our results show remarkable bias of ERA5L soil temperature in winter that likely correlates with snow depth (Figure 2).*"
Table 4: Are there any snow density measurements from the site that could clarify which one of the models is right (or if both are wrong).

There's no snow measurements used here. As we've stated "we do not imply that GEOtop-based simulations are correct or representing metamorphism in Arctic snow accurately, they demonstrate that simulations with snow cover of similar mass but different density are able to match ground-temperature observations far better than ERA5L.". In fact, simulating critical snow physical variables in Arctic is challenging (see Domine et al., 2019).

I. 152: Make it clear that this is "ERA5L PF extent as defined in this study", it is clear that the shallow soil column makes it very difficult to relate this to "true PF extent change". Such statements can easily be misunderstood, compare to "Lawrence, D.M. and Slater, A.G., 2005. A projection of severe near-surface permafrost degradation during the 21st century. Geophysical Research Letters, 32(24)." and the resulting comment by Burn & Nelson. This issue is again explained later in the discussion, but make it clear already here, that this by no means represents real PF extent changes. It will be changed as "Near-surface permafrost area of ERA5L as defined in this study decreased with a rate of -0.11 (-0.08) × 106 km2 year-1 based on hourly (annually) soil temperature (Figure 6)."

I. 168: what do you mean by "although less permafrost processes are coupled"? Compared to CLASS-CTEM presented by Melton et al., (2019), HTESSEL includes less physical processes regarding permafrost. We'll change this part to "...although fewer permafrost specific processes are included in the HTESSEL..." to clarify.

L. 170: When I look at Fig. 5, I don't quite understand why there is a "remarkably low
bias in PF extent". Your explanations later seem to go in the direction that this might rather be a coincidence, since biases in different regions cancel each other?

The low bias of ERA5L summary statistics in Table 2 is a coincidence as the warm bias in high latitudes (Canada and Alaska) and cold bias in mid-low latitudes canceled each other (Figure 3). The "remarkably low bias in permafrost area" is because 1) ERA5L can only represent the "near-surface" permafrost area due to the shallow soil column; 2) warm bias of soil temperature in high latitudes, especially in northern Canada and Alaska (Figure 1).

Furthermore, ERA5L cannot really represent the discontinuous and continuous permafrost zones, so fractional PF coverage is by definition not included.

The 50% permafrost coverage is used for the IPA map regarding continuous and discontinuous permafrost. Details are present in Section 2.4–"By following Melton et al., 2019, a threshold of 50% (continuous and discontinuous permafrost zones) for the IPA map and 0.5 for the PZI map, respectively, are used for permafrost area estimation and for comparing areas with binary maps."

Sect. 5.4: Dedicated snow models like CROCUS and Snowpack also include formulations for compaction due to wind drift which likely occurs at LdG(?). If I understand correctly, this is neither included in the ERA5L model nor in GEOtop? This should be stated, especially since there seem to be no field measurements of snow densities from the site which could clarify which model is more right? I would certainly agree that the GEOtop snow densities look much more realistic, but that's more an educated opinion, rather than science.

The snow compaction due to wind effects is represented in GEOtop (2.0) following Pomeroy et al., (1993), while not in the ERA5L. We considered the wind compaction for all terrain types in LdG except the tall shrubs site. In section 3.2, we'll add "Snow compaction due to wind effects is considered in 1-D for all terrain types except the tall shrub site (Pomeroy et al., 1993)."to clarify. In addition, we'll change the last sentence to "An additional contribution of GEOtop to higher snow densities in tundra

TCD
**environments may be considering the effect of blowing snow (cf, Pomeroy et al., 1993)" to clarify.**

Discussion general: Consider adding a Section "Implications" or similar – especially the findings on the snow cover and the shallowness of the ground representation are very interesting also for improvements of further reanalysis generations. To me it almost looks like that one might get a pretty good performance for permafrost parameters by doing a couple of obvious improvements of the ground and snow models (which likely wouldn't even cost a lot of additional computation). You study is a great reference for this, and stating this clearer will likely increase the impact of the paper.

The implications were given at the end of the manuscript as part of the conclusions. In the revision, we'll move this part to the new Section 5.5 Implications (as below) in order to make the manuscript more readable:

"While global reanalyses provide urgently needed meteorological drivers for permafrost simulation, their soil data is not well suited for directly informing permafrost research or local adaptation decisions. As such, simulations using permafrost-specific land-surface models driven by reanalyses (Cao et al., 2019a, Fiddes2015) will likely gain importance. Making future soil-temperature products like ERA5L directly usable will require significant permafrost-specific alterations, especially snow cover and the shallowness of the ground representation, to the land-surface models used. If indeed the value of the parameter  $c_{\xi}$  in the snow metamorphism of HTESSL is in error, then this would be an easy improvement."

**References**

Cao, B., Quan, X., Brown, N., Stewart-Jones, E., and Gruber, S.: GlobSim (v1.0): deriving meteorological time series for point locations from multiple global reanalyses, Geosci. Model Dev., 12, 4661–4679, https://doi.org/10.5194/gmd-12-4661-2019,
2019.

Domine, F., Picard, G., Morin, S., Barrere, M., Madore, J. B., and Langlois, A.: Major Issues in Simulating Some Arctic Snowpack Prop- erties Using Current Detailed Snow Physics Models: Consequences for the Thermal Regime and Water Budget of Permafrost, Journal of Advances in Modeling Earth Systems, 11, 34–44, https://doi.org/10.1029/2018MS001445, 2019.

Karjalainen, O., Aalto, J., Luoto, M., Westermann, S., Romanovsky, V. E., Nelson, F. E., Etzelmüller, B., and Hjort, J.: Data descrip- tor: Circumpolar permafrost maps and geohazard indices for near-future infrastructure risk assessments, Scientific Data, 6, 1–16, https://doi.org/10.1038/sdata.2019.37, 2019a.

Melton, J. R., Verseghy, D. L., Sospedra-Alfonso, R., and Gruber, S.: Improving permafrost physics in the coupled Canadian Land Surface Scheme (v.3.6.2) and Canadian Terrestrial Ecosystem Model (v.2.1) (CLASS-CTEM), Geoscientific Model Development, 12, 4443–4467, https://doi.org/10.5194/gmd-12-4443-2019, 2019.

Obu, J., Westermann, S., Bartsch, A., Berdnikov, N., Christiansen, H. H., Dashtseren, A., Delaloye, R., Elberling, B., Etzelmüller, B., Kholodov, A., Khomutov, A., Kääb, A., Leibman, M. O., Lewkowicz, A. G., Panda, S. K., Romanovsky, V., Way, R. G., Westergaard-Nielsen, A., Wu, T., Yamkhin, J., and Zou, D.: Northern Hemisphere permafrost map based on TTOP modelling for 2000–2016 at 1 km2 scale, Earth-Science Reviews, 193, 299–316, https://doi.org/10.1016/j.earscirev.2019.04.023, 2019.

Pomeroy, J. W., Gray, D. M., and Landine, P. G.: The Prairie Blowing Snow Model: characteristics, validation, operation, Journal of Hydrology, 144, 165–192,

TCD
https://doi.org/10.1016/0022-1694(93)90171-5, 1993.

---

## Author Comment (AC2) · 12 Jun 2020

**Responses of Anonymous Referee #2**

The authors would like to thank the reviewer for the constructive feedback, and the thorough assessment of the manuscript. Below we provide a point-to-point response to each comment, reviewer comments are given in black, responses are given in blue. Additionally, we have included details of how we intend to address these changes in a revised submission.

**General comments:**

[Figure]

This paper presented a good assessment of the soil temperature at a large scale using in-situ observations and previous products/maps. Understanding current soil temperature bias in reanalysis could improve further Earth-system model design by accounting more essential permafrost processes and hence benefit the permafrost community. This paper is generally well written. I have some comments for further revisions.

**Major comments:**

- As Reviewer#1 stated, some important points became clear a little bit late. To casual readers, this may be not easy to follow.
  Please see our responses to the general comments of RC#1.

- The authors MUST recheck this statement in L70–71. From the ERA5L website, they said: "Temperature of the soil in layer 1 (0–7 cm) of the ECMWF Integrated Forecasting System. The surface is at 0 cm. Soil temperature is set at the middle of each layer, and heat transfer is calculated at the interfaces between them." This is very important because these depths were used to interpolate soil temperature profiles and to determine ALT, if my guess is correct. If incorrect depths were used, the comparisons were already artificially altered.
  We've noticed the differences of soil depth from the ERA5L document website and model description document (see Table 8.7 in IFS Documentation CY45R1 (https://www.ecmwf.int/en/elibrary/18714-part-iv-physical-processes). We also contacted the scientist from ECMWF, and I simply copied the reply below.

  *"The soil temperature of a given layer is an averaged value over of the thickness of that layer and assigned to the middle of the layer. From the modeling point of*

*view this temperature is a valid temperature for any point in the layer, whereas
in reality it'll be different depending on the depth. This is one of the limitations
when the soil is discretised in a finite number of layers."*

For this reason, we followed the depth in model document as described in L70–
71: *"The soil column of ERA5L is discretized into four layers with node depths
(layer boundaries) of 0.07 (0–0.07), 0.21 (0.07–0.28), 0.72 (0.28–1.00), and 1.89
(1.00–2.89) m."*

- The authors should describe the estimate of ALT by using ERA5L.
  In paragraph of section 3.1, we'll add *"ERA5L ALT is derived through linear inter-
  polation from ERA5L soil temperature-depth profiles."*

- Did the authors consider the uncertainties from vegetation?
  Our results indicated that the ERA5L soil temperature bias are mainly from the
  MAAT bias (Figure 1), and snow (see larger bias in winter from Figure 2, and Fig-
  ure 7). That's why we considered the MAAT bias and snow as possible predictors
  rather than the vegetation, and the linear model of Eq. (1) indicated the success
  of variable selection. We hope you agree.

- In section 2.3, I miss a description of air temperature observation, while it is used
  for analyses of ERA5L soil temperature bias (i.e. in Table 1 and the linear model).
  Authors have to add a brief description here, and even show them in a proper way.
  This could be easily done, for example, by changing the shape of the station with
  both air and soil temperatures in Figure A1.
  In the revision, we'll add the air temperature observation info (see the attached
  figure).

**Specific comments:**
P2, L27: The RMSE of reanalyses soil temperature? Please clarify.
Will revised as *"For example, over the Qinghai–Tibetan Plateau (QTP), Hu et al.,(2019); Yang and Zhang (2018) reported that the root mean squared error (RMSE) of daily soil temperature from different reanalyses ( i.e. ERA5-Interim/Land, MERRA-2, and CFSR) was up to 1.8–5.1∘C, and they are generally consistently cold bias."*

P2, L40: ... and example numerical or process-based simulation...
Will be revised.

P2, L57: Note that ERA5L is now available from 1981.
In the revision, we'll add *"Note that, during writing only ERA5L data after 2001 are released and hence this evaluation is conducted for data between 2001–2018."*.

P4, L86: The soil temperature from the TTOP and CP maps are used as comparisons, please as mention here.
In the revision, we'll add *"The mean annual ground temperatures (MAGT) from the TTOP and CP maps are also used for ERA5L evaluation."*

P4, L89: ...(denoted as PZI map)","..., should it be ";"? Similar in L91.
Will be revised.

P4, L97: The MAGT of TTOP and CP maps are additionally used as reference in your Table 1 and Figure 3. Please clarify here.
Will be revised.

P5, L104: ...in the same ERA5L grid cell...
Will be revised.

P5, L107: ...of ERA5L soil temperature....
Will be revised.

P5, L126: there is a repeat of the "the".
Will be revised.

P5, L134: ...and (2) an increase of 1 m wSD$_{max}$
Will be revised.

P7, L149: Is the ALT also overestimated in high latitudes and underestimated in high altitudes?
It is difficult to say as most of the sites in mid-low are excluded before evaluation since their ALT $>$ 1.89 m. In this case, the evaluation shown here are generally for high latitudes (see Figure 5 for the site distribution of ALT $<$ 1.89 m). We'll change the caption of Figure 4 to *"The observed sites are mainly located in high latitudes, and the distribution is present in Figure 5."* to clarify.

P10, L164: Also mention the high spatial (and maybe temporal) resolution here, this is one of the most significant features of ERA5 compared to the others.
Will be revised as *"ERA5L has a number of advantages, such as long-term (back to 1950, eventually), high spatial resolution, and global coverage."*

P13, L215: ...for $c_\xi$ in Eq. B5...
Will be revised.

P13, L216: It should be 150 kg m$^{-3}$ based on Eq. B5, please double check.
Will be revised to 150 kg m$^{-3}$.

P14, L236: Underestimate permafrost...(what)? Permafrost area? Please clarify.
It is permafrost area, and will be clarified in the revision.

P14, L252–253: The bracket is incomplete
Will be revised.

P14, L255: Brackets are needed here for the url.
Will be revised.

P15, L270: Add space between m and s$^{-2}$
Will be added.

P16, L278: $\rho_\xi$ is not included in Eq. (B5).
The sentence will be changed to *"where the $a_\xi$, $b_\xi$, and $c_\xi$ are constant values of $2.8 \times 10^{-6}$ ($s^{-1}$), 0.042 (−) and 460 ($m^3$ $kg^{-1}$) derived or modified from Anderson(1976) and Jordan et al. (1999)."*

P16, L280: Considering move $\Delta\beta_s = 0$ to the upper so that Eq B6 would be aliened with the state of Eq. B8 and B10
Will be revised.

P16, L297: ...ice density of 920...
Will be revised.

**Specific comments:**

- Table 1: This is only for the observations in permafrost regions. Please clarify in the caption otherwise including the observations in non-permafrost regions.
  Yes, this is only permafrost regions. The caption is changed to *"Summary of soil temperature observations in permafrost regions..."*

- Figure 3: In the caption, it should be "...(observation-ERA5L)..."
  Will be revised.

- Figure 6: Considering add unit to the permafrost area changing rate.
  Unit will be added in the caption–*"...Linear lines represent the trend of permafrost area ($10^6$ $km^2$ $year^{-1}$) based on linear model..."*

[Figure]

[Figure]

Legend:

○ MAAT    △ No MAAT

| | |
|---|---|
| CMA | GTN-P |
| WDC | NPS |
| Nordicana D | USGS |
| GI-UAF | HiWATER |
| Tibet-OBS | Others |
| CTP-SMTMN | NPF |

**Fig. 1.**

---

## Author Comment (AC3) · 12 Jun 2020

**Responses of Anonymous Referee #3**

The authors would like to thank the reviewer for the constructive feedback, and the thorough assessment of the manuscript. Below we provide a point-to-point response to each comment, reviewer comments are given in black, responses are given in blue. Additionally, we have included details of how we intend to address these changes in a revised submission.

This paper assesses the utility of ERA5L soil temperature products for permafrost

studies by using a wide range of global station data from both permafrost and non permafrost regions as well as detailed simulation experiments at a specific site. The authors find that ERA5L has large biases making the product problematic for permafrost studies. This study is a valuable contribution as we increasingly use reanalysis products for land surface modeling studies, especially at regional or global scales and insights into performance of these products are useful. Additionally, such studies may help to guide future developments in land surface schemes used in reanalyses. I recommend publishing after considering my (mainly minor) comments. (in grammatical comments changes are CAPITALIZED)

The manuscript will be carefully edited by native speaker with strong permafrost background in order to improve the language.

1. l.3 "is predicted TO BE too warm...."
2. l.19 "Reanalysis, ASSIMILATES"
9. l.74 "These INCLUDE"
13. l129 "A linear model..."
16. l.147 "While ERA5L does not have DATA allowing deep ALT values to be computed"
26, l.232. use of "low" here is confusing. you are biased to low densities, you do not have a low bias. I would say "a low-density bias" to make it clear.

Will be edited by native speaker

3. l.28 what is ERA5-Interim/Land? Seems a confusion of the products
It was a typo, should be ERA-Interim/Land.

4. l.29 "consistently cold BIASED."
Will be revised to *"..,and generally the soil temperature from different reanalyses (i.e. ERA5-Interim/Land, MERRA-2, and CFSR)* **shows consistently cold bias."**

5. l.54 I think the HTESSEL ref could do with a publication citation.
The latest ERA5 paper, Hersbach et al., (2020), that describes HTESSEL is added here.

6. l.57 now available from 1981.
In the revision, we'll add *"Note that, during writing only ERA5L data after 2001 are released andhence this evaluation is conducted for data between 2001–2018.".*

7. Section 2.2.1 what do B1 and B2 refer to?
Will be revised to *"Appendix B1"* and *"Appendix B2"* to clarify.

8. l.71 is the node really at the lower boundary (0.07) in soil layer 1?
Reviewer #2 also mentioned this issue. We copied the responses here.

Based on the Table 8.7 in IFS Documentation CY45R1 (https://www.ecmwf.int/en/elibrary/18714-part-iv-physical-processes), the lower boundary is 0.07 m, although this is different from the description in ERA5L document website). We also contacted the scientist from ECMWF, and I simply copied the reply below.

*"The soil temperature of a given layer is an averaged value over of the thickness of that layer and assigned to the middle of the layer. From the modeling point of view this temperature is a valid temperature for any point in the layer, whereas in reality it'll be different depending on the depth. This is one of the limitations when the soil is discretised in a finite number of layers."*

For this reason, we followed the depth in model document as described in L70–71:
*"The soil column of ERA5L is discretized into four layers with node depths (layer boundaries) of 0.07 (0–0.07), 0.21 (0.07–0.28), 0.72 (0.28–1.00), and 1.89 (1.00–2.89) m."*

10. Section 2.3 and Table 1 are all stations boreholes? If so perhaps explicitly state that.
No, some sites are from boreholes, e.g., GTN-P, but sites like CMA, HiWATER, are from soil temperature sensor of meteorological stations. In Section 2.3, we'll change the first sentence to *"Soil temperatures from 639 borehole sites or meteorological stations located in permafrost regions are used"* to clarify.

11. l.90-91 and driven by ERA-Interim air temperature.
The TTOP map compiled by Obu et al., (2019) was driven mainly by MODIS LST, but the data gaps due to cloud cover was filled by downscaled ERA-Interim air temperature. Please see Section 2.2 and Figure 1 from Obu et al., (2019). In the revision, we'll change the sentence to
*"...(3) the 1-km Northern Hemisphere permafrost map Obu et al., (2019) based on the semi-physical Temperature at the Top Of the Permafrost table (TTOP) model (TTOP map) driven by Moderate Resolution Imaging Spectroradiometers (MODIS) land surface temperature that filled by downscaled ERA-Interim air temperature;"* to clarify.

12. l111-114: I don't quite understand the motivation for the two definitions of near surface permafrost I think a sentence explaining why you do this would be helpful for the reader.

The two algorithms are defined here to derived ERA5L soil temperature-based permafrost area. In the revision, this sentence will be changed to *"To evaluate the ERA5L near-surface permafrost area, permafrost is diagnosed from ERA5L soil temperature in two ways..."*

14. l.137 What depth are these MAGT's? Averaged across time or space? Please provide a bit more detail here.
Reviewer #1 also had similar comment. We copied the response here as well.

In Section 3.1, we'll add *"The TTOP and CP map are derived using equilibrium model, and MAGT is given as an average of the entire period (MAGTavg), i.e. 2002–2014 for the CP map and 2002–2016 for the TTOP map, without uniform/specific soil depth. For better evaluation purpose, we aggregate all available observed MAGTs during the period by averaging, and then compared against the MAGTavg of these two maps. Note that the performance of CP and TTOP maps may be lower here than reported in the original publications due to differing observations (depths, periods and proportion of sites in mountains) used."* to clarify

15. l.143 more prevalent snow and soil freezing in the model or in reality? Please clarify. If in reality, then permafrost regions do not necessarily have more prevalent snow than non-permafrost regions.
This is in the HTESSEL model or ERA5L based on the bias comparison of in permafrost and non-permafrost regions(Table 2; Figure 1). We will revise this part to *"In addition to the worse performance of MAAT in these regions, the result suggests that more prevalent snow and soil freezing may reduce the suitability of HTESSEL for soil temperature simulation."*

17. l.153 "(annually)" is it an annual average? Please clarify.
Yes, it will be revised to *"Near-surface permafrost area of ERA5L as defined in this study decreased with a rate of -0.11 (-0.08) $\times 10^6$ km$^2$ year$^{-1}$ based on hourly (annually) mean soil temperature (Figure 6)."* to clarify.

18 Figure 3 Interesting latitudinal trend in c,d. Can you shed more light on this in the discussion? I guess densification processes at high latitudes (badly represented wind?) What is driving the cold bias at low latitudes?
As Reviewer #1 mentioned, HTESSEL does not have a representation of wind effects on snow densification. In this case, blowing snow. Both Figure 1 and the linear model (Eq. 1) indicated that the cold bias at low latitudes is largely due to the MAAT bias. In Section 5.1, we will add
*"Our results indicate that the cold bias of ERA5L in mid-low latitudes is highly aligned with the MAAT bias (Figure 1), and this could also be reflected by the linear model (Eq. 2)."*

19. Figure 4 perhaps add the mean value that you cite in the text here.
Will be revised as attached.

20. l.170 "shows REMARKABLY"
Will be revised.

21. l.194 "Even for A"
Will be revised.

22. l.198 "This issue is KNOWN"
Will be revised.

23. l.208 "as AN exponential..."
Will be revised.

24. l.226 soil temperatureS MATCH..."
Will be revised as suggested.

25. l.230 But what about the cold bias you see? the bias appears to evenly spread (figure3) why does this not give a similar spread in ALT estimates (Figure 4) and a related underestimation of ALT?
This is because the shallow ALT sites ($<$ 1.89 m) are mainly in high latitudes (Figrue 5), and in high latitudes the soil temperature was found too warm. This is aligned with Figure 3. In the revision, we'll change the caption of Figure 4 to *"The observed sites are mainly located in high latitudes, and the distribution is present in Figure 5." to clarify.*

**References**

Obu, J., Westermann, S., Bartsch, A., Berdnikov, N., Christiansen, H. H., Dashtseren, A., Delaloye, R., Elberling, B., Etzelmüller, B., Kholodov, A., Khomutov, A., Kääb, A., Leibman, M. O., Lewkowicz, A. G., Panda, S. K., Romanovsky, V., Way, R. G., Westergaard-Nielsen, A., Wu, T., Yamkhin, J., and Zou, D.: Northern Hemisphere permafrost map based on TTOP modelling for 2000–2016 at 1 km2 scale, Earth-Science Reviews, 193, 299–316, https://doi.org/10.1016/j.earscirev.2019.04.023, 2019.

Hersbach, H., Bell, B., Berrisford, P., Hirahara, S., Horányi, A., Muñoz-Sabater, J., Nicolas, J., Peubey, C., Radu, R., Schepers, D., Simmons, A., Soci, C., Abdalla, S., Abellan, X., Balsamo, 80 G., Bechtold, P., Biavati, G., Bidlot, J., Bonavita, M., Chiara, G., Dahlgren, P., Dee, D., Diamantakis, M., Dragani, R., Flemming, J., Forbes, R., Fuentes, M., Geer, A., Haimberger, L., Healy, S., Hogan, R. J., Hólm, E., Janisková, M., Keeley, S., Laloyaux, P., Lopez, P., Lupu, C., Radnoti, G., Rosnay, P., Rozum, I., Vam- 85 borg, F., Villaume, S., and Thépaut, J.-N.: The ERA5 Global Reanalysis, Quarterly Journal of the Royal Meteorological Society, n/a, qj.3803, https://doi.org/10.1002/qj.3803, 2020.

[Figure]

ERA5L ALT (m)

Observed ALT (m)

mean ALT (m)
ERA5L$_{avg}$ = 1.67
OBS$_{avg}$ = 0.82

N = 79 (106, 787)
wBIAS = 0.85
wMAE = 0.88
wRMSE = 0.98

**Fig. 1.**

---

## Author Response (AR1)

The authors would like to thank the reviewer and editor for their constructive feedback, and the thorough assessment of the manuscript. Below we provide a point-to-point response to each comment, reviewer comments are given in black, responses are given in blue.

**Responses of Anonymous Referee #1**

The study "The ERA5-Land Soil-Temperature Bias in Permafrost Regions" by Cao et al. evaluates the performance of the ERA5L reanalysis for ground temperatures and other ground-temperature-related parameters in permafrost areas. Although ground temperature is not a main target parameter for such reanalysis products, the study will be a valuable scientific contribution and I recommend publication after carefully revising the manuscript.

**Major Comment/Recommendation:**

When reading through the manuscript, many important points only became clear to me very late, i.e. in the Discussion. The temperature comparisons of the different products in the Results section, for example, left me wondering on the interpretation and implications. The same applies to the findings on the sizable reduction of "permafrost area" in ERA5L, which only much later is resolved as likely being more an artefact of the model than reality. To a casual reader, the manuscript appears to make a number of potentially bold statements, without providing any hint that the interpretations/ clarification of implications are provided at some later stage in the Discussion (where some casual readers might miss it). While the strict separation of the different manuscript parts is in line with accepted methodology for scientific writing, I recommend guiding the reader through the manuscript in a better way. I have made more specific annotations and suggestions under general comments.

We agree, hints are added as suggested in the specific comments. Especially, Section 5.2 is mentioned in the caption of Table 2 in order to avoid any possible misunderstanding. We moved the implication part from Section Conclusions to the Discussion (Section 5.5).

**General comments:**

Sect. 2.2 Remind the reader in one sentence what HTESSEL is, this is somewhat hidden in the previous text. We changed this part to "... A more realistic representation of snow is used in the ERA5 land surface model compared with its predecessor, ERA-Interim."

Sect. 2.3 and 3.1: Please add information on the depths of the available borehole temperatures and how this compares to the shallow ground representation in ERA5L. The Biskaborn-data set, for example, contains many borehole measurements at much deeper layers than ERA5L can represent, so (how) are these measurements used?

Only the observed temperature within the ERA5L soil temperature column, i.e. 0–2.89 m, were used here. In Table 1, we added the depth range of used soil temperature observations for each data source. In Section 3.1, we revised as "For the purposes of evaluation, temperature observations were only used from depths between 0 m and 2.89 m, corresponding to the range of the ERA5L soil column. Temperature values were grouped according to their depth into one of the ERA5L soil layers."

1.112: the first criterion is unclear, is this "if T of any of the four layers is constantly below zero for two years"?

Yes. It is changed as "An ERA5L grid cell is considered to be underlain by permafrost if either of the following conditions are true: (1) soil temperature in any of the four soil layers has an hourly temperature below  $0^{\circ}C$  for two consecutive years (ERA5LH)"

Sect. 3.2 The added value of this is unclear at this stage of the manuscript, it seems to be rather unrelated to the main purpose, i.e. compare the direct ground T output of ERA5L to observations. This becomes clear only much later, but please add a few sentences on the purpose already here.

At the beginning of Section 3.2, we added "Our results show remarkable bias of ERA5L soil temperature in winter that likely correlates with snow depth (Figure 2). For this reason, the suitability of ERA5L soil temperature and the effect of snow-density bias are further investigated with a site specific simulation example at a densely instrumented location near Lac de Gras (LdG), N.W.T., Canada (Figure 1A). This detailed permafrost simulation example provides an opportunity to evaluate ERA5L soil temperature under different terrain (e.g. vegetation, soil properties) and snow conditions." Table 2+3: add the references to the different products used (at least in the caption), so that the readers don't have to search for the abbreviations in the text.

The reference is added in the caption:

Table 2: "...The  $MAGT_{avg}$  is the average MAGT: 2001–2018 for ERA5L, 2000–2014 for the CP map (Karjalainen et al., 2019), and 2002–2016 for the TTOP map (Obu et al., 2019)."

Table 3: "Note that the CP map only represents permafrost distribution north of 30° N (Karjalainen et al., 2019), TTOP map represents permafrost distribution of the Northern Hemisphere (Obu et al., 2019), and the others represent the area of north of 60° S. Permafrost area from literature is given with their definition in this study.

1. 129: the purpose of the equation is unclear, and must be explained in more detail. To clarify the purpose, we conducted revision below.

- 1) refer to Eq. 1 in Section 3.1: "MAAT bias and maximum snow depth  $(SD_{max})$  were selected as candidate variables to be assessed as possible predictors of ERA5L soil temperature bias (see Eq. 1)".
- 2) this sentence is changed to "The following linear model was used to predict ERA5L soil temperature bias in permafrost regions using MAAT bias and snow depth as predictor variables.".

If I understand correctly, you relate the bias in MAGT to the bias in MAAT, using the snow depth (which has no bias, I guess since measurements are not available?).

Yes, snow measurements are not available.

Does the intercept of 0.15 make sense, i.e. zero bias in MAAT and zero snow produces an MAGT bias of 0.15? Should one not rather prescribe an intercept of 0 in the equation? I guess it would not change much, considering the R2 of 0.47 of the relationship.

We can expect uncertainty of the linear model with  $R^2$  of 0.47 since it was fitted with limited observations, i.e. 239 grid cells. However, the intercept of 0.15 makes sense. It means even no MAAT bias and snow cover is present, ERA5L soil temperature in permafrost regions could still have bias that may from the other variables, i.e. due to the mismatched depth of observations and ERA5L soil layer.

Table 2: I assume the comparison is done for the individual years when- and wherever an entire year of observations is available?

Yes, for MAAT, SO, and MAGT evaluation, the comparison is done for individual available years, while the  $MAGT_{avg}$  is the average MAGT for the entire long period. In the caption, we added "MAAT, SO, and MAGT were evaluated for each individual year, while  $MAGT_{avg}$  was carried through once for the entire period and are based on sparse data."

How does this relate to CP and TTOP which represent longer periods, are only observation that span the entire periods used? If not, to what extent does the availability of observations influence these comparisons - many observations are likely taken in recent years, which on average were warmer than earlier periods. There is the passage starting with "Note that the performance of CP and TTOP maps may be lower here than reported in: : :", but the implication of this is not really clear.

In Section 3.1, we added "The TTOP and CP map were derived using an equilibrium model, and MAGT is given as an average of the entire period ( $MAGT_{avg}$ ). This corresponds to 2002–2014 for the CP map and 2002–2016 for the TTOP map, without uniform/specific soil depth. To better evaluate, we aggregated all available observed MAGTs during the period by averaging, and then compared against the MAGTavg of these two maps. Note that the performance of CP and TTOP maps may be lower here than reported in the original publications due to the fact that we evaluate them with a different set of observations (different depths, periods and proportion of sites in mountains)." to clarify. The sentence of "Note that the performance of CP and TTOP maps may be lower here than the performance of CP and TTOP maps may be lower here than the performance of CP and TTOP maps may be lower here that the performance of CP and TTOP maps may be lower here that the performance of CP and TTOP maps may be lower here that the performance of CP and TTOP maps may be lower here that the performance of CP and TTOP maps may be lower here that the performance of CP and TTOP maps may be lower here that the performance of CP and TTOP maps may be lower here that the performance of CP and TTOP maps may be lower here than reported in..." is removed from Section 4.1

Table 2 seems to suggest that ERA5L is considerably better than CP and TTOP for PF areas, but it is unclear if that conclusion can indeed be drawn. This is not only considering the study periods, but also the spatial distribution of the measurement sites (heavily biased towards China, SE Russia and Alaska according to Fig. 2). This point is adequately discussed in 5.2, but it would be good if some of it could be mentioned already here. At least include a statement "see Sect. 5.2 for a detailed discussion" in the text.

Yes, the summary statistics with sparse data would be misleading. In the revision, we added " $MAGT_{avg}$  must be interpreted cautiously, taking into consideration the points outlined in Section 5.2." in the caption of Table 2.

l. 137: typo "bilinearly" Revised.

Fig. 1: add units in the figure. The unit, °C, is now included in the legend.

Fig. 2 is only presented in one sentence in the text. This should be presented in more detail. I suggest using this to motivate Section 4.3 (see also comment above).

Figure 2 is added in Section 3.2: "Our results show remarkable bias of ERA5L soil temperature in winter that is thought to correlate with snow depth (Figure 2)."

Table 4: Are there any snow density measurements from the site that could clarify which one of the models is right (or if both are wrong).

There's no snow measurements used here. As we've stated "While we do not imply that the GEOtop simulations are correct or accurately represent metamorphism in Arctic snow (see Domine et al., 2019), they do demonstrate that simulations with snow cover of similar mass but different density are able to match groundtemperature observations far better than ERA5L.". In fact, simulating critical snow physical variables in Arctic is challenging (see Domine et al., 2019).

1. 152: Make it clear that this is "ERA5L PF extent as defined in this study", it is clear that the shallow soil column makes it very difficult to relate this to "true PF extent change". Such statements can easily be misunderstood, compare to "Lawrence, D.M. and Slater, A.G., 2005. A projection of severe near-surface permafrost degradation during the 21st century. Geophysical Research Letters, 32(24)." and the resulting comment by Burn & Nelson. This issue is again explained later in the discussion, but make it clear already here, that this by no means represents real PF extent changes.

It is changed as "Near-surface permafrost area of ERA5L as defined in this study decreased at a rate of -0.11 (-0.08)  $\times 10^6$  km2 year-1 based on hourly (annual) mean soil temperature."

1. 168: what do you mean by "although less permafrost processes are coupled"?

Compared to CLASS-CTEM presented by Melton et al., (2019), HTESSEL includes less physical processes regarding permafrost. We changed this part to "Compared to a coarse-grid ( $\sim 2.8^{\circ}$ ) simulation (Figure 4 from Melton et al., 2019), ERA5L often has more reasonable results in its deepest soil layer, despite the fact that fewer permafrost-specific physics are included in the HTESSEL." to clarify.

L. 170: When I look at Fig. 5, I don't quite understand why there is a "remarkably low bias in PF extent". Your explanations later seem to go in the direction that this might rather be a coincidence, since biases in different regions cancel each other?

The low bias of ERA5L summary statistics in Table 2 is a coincidence as the warm bias in high latitudes (Canada and Alaska) and cold bias in mid-low latitudes canceled each other (Figure 3). The "remarkably low bias in permafrost area" is because 1) ERA5L can only represent the "near-surface" permafrost area due to the shallow soil column; 2) warm bias of soil temperature in high latitudes, especially in northern Canada and Alaska (Figure 1).

Furthermore, ERA5L cannot really represent the discontinuous and continuous permafrost zones, so fractional PF coverage is by definition not included.

The 50% permafrost coverage is used for the IPA map regarding continuous and discontinuous permafrost. Details are present in Section 2.4: "Following Melton et al., 2019, we apply a threshold of 50% (corresponding to the continuous and discontinuous permafrost zones) and 0.5 for the IPA map and the PZI map, respectively, to allow for meaningful comparison with the other maps."

Sect. 5.4: Dedicated snow models like CROCUS and Snowpack also include formulations for compaction due to wind drift which likely occurs at LdG(?). If I understand correctly, this is neither included in the ERA5L model nor in GEOtop? This should be stated, especially since there seem to be no field measurements of snow densities from the site which could clarify which model is more right? I would certainly agree that the GEOtop snow densities look much more realistic, but that's more an educated opinion, rather than science.

The snow compaction due to wind effects is represented in GEOtop (2.0) following Pomeroy et al., (1993), while not in the ERA5L. We considered the wind compaction for all terrain types in LdG except the tall shrubs site. In section 3.2, we added "Snow compaction due to wind effects is considered in 1-D for all terrain types except for the tall shrub site (Pomeroy et al., 1993)."to clarify. In addition, we changed the last sentence to "An additional contribution of GEOtop to higher snow densities in tundra environments may be

**the effect of blowing snow (cf, Pomeroy et al., 1993)" to clarify.**

Discussion general: Consider adding a Section "Implications" or similar - especially the findings on the snow cover and the shallowness of the ground representation are very interesting also for improvements of further reanalysis generations. To me it almost looks like that one might get a pretty good performance for permafrost parameters by doing a couple of obvious improvements of the ground and snow models (which likely wouldn't even cost a lot of additional computation). You study is a great reference for this, and stating this clearer will likely increase the impact of the paper.

The implications was given at the end of the manuscript as part of the conclusions. We now moved this part to the new Section 5.5 Implications (as below) in order to make the manuscript more readable:

"While global reanalyses provide urgently needed meteorological drivers for permafrost simulation, their soil data is not well suited for directly informing permafrost research or local adaptation decisions. As such, simulations using permafrost-specific land-surface models driven by reanalyses (Cao et al., 2019a, Fiddes2015) will likely be increasingly important in the provision of permafrost climate services. Making future soil-temperature products like ERA5L directly usable will require significant permafrost-specific alterations in model design, especially with respect to snow cover and the total depth of the ground representation for the land-surface models that are used. If indeed the value of the parameter  $c_{\overline{z}}$  in the snow metamorphism of HTESSL is in error, then this would be an easy improvement."

**Responses of Anonymous Referee #2**

**General comments:**

This paper presented a good assessment of the soil temperature at a large scale using in-situ observations and previous products/maps. Understanding current soil temperature bias in reanalysis could improve further Earth-system model design by accounting more essential permafrost processes and hence benefit the permafrost community. This paper is generally well written. I have some comments for further revisions.

**Major comments:**

- As Reviewer#1 stated, some important points became clear a little bit late. To casual readers, this may be not easy to follow.

Please see our responses to the general comments of RC#1.

Figure 1: Distribution of soil temperature stations. Stations in permafrost regions are in color while the gray ones are non-permafrost (NPF) stations. Stations in circle additional has air temperature observation while the triangle ones do not.

- The authors MUST recheck this statement in L70–71. From the ERA5L website, they said: "Temperature of the soil in layer 1 (0-7 cm) of the ECMWF Integrated Forecasting System. The surface is at 0 cm. Soil temperature is set at the middle of each layer, and heat transfer is calculated at the interfaces between them." This is very important because these depths were used to interpolate soil temperature profiles and to determine ALT, if my guess is correct. If incorrect depths were used, the comparisons were already artificially altered.

We've noticed the differences of soil depth from the ERA5L document website and model description document (see Table 8.7 in IFS Documentation CY45R1

(https://www.ecmwf.int/en/elibrary/18714-part-iv-physical-processes). We also contacted the scientist from ECMWF, and I simply copied the reply below.

"The soil temperature of a given layer is an averaged value over of the thickness of that layer and assigned to the middle of the layer. From the modeling point of view this temperature is a valid temperature for any point in the layer, whereas in reality it'll be different depending on the depth. This is one of the limitations when the soil is discretised in a finite number of layers."

For this reason, we followed the depth in model document as described in L70-71: "The ERA5L soil column is discretized into four layers with node depths (layer boundaries) at 0.07 (0-0.07), 0.21 (0.07-0.28), 0.72 (0.28-1.00), and 1.89 (1.00-2.89) m"

- The authors should describe the estimate of ALT by using ERA5L. In paragraph of section 3.1, we added "ERA5L ALT was derived by linearly interpolating the ERA5L soil temperature-depth profiles."
- Did the authors consider the uncertainties from vegetation? Our results indicated that the ERA5L soil temperature bias are mainly from the MAAT bias (Figure 1), and snow (see larger bias in winter from Figure 2, and Figure 6). That's why we considered the MAAT bias and snow as possible predictors rather than the vegetation, and the linear model of Eq. (1) indicated the success of variable selection. We hope you agree.
- In section 2.3, I miss a description of air temperature observation, while it is used for analyses of ERA5L soil temperature bias (i.e. in Table 1 and the linear model). Authors have to add a brief description here, and even show them in a proper way. This could be easily done, for example, by changing the shape of the station with both air and soil temperatures in Figure A1.

In the revision, we added the air temperature observation info to Figure A1.

**Specific comments:**

P2, L27: The RMSE of reanalyses soil temperature? Please clarify.

Revised as "For example, over the Qinghai–Tibetan Plateau (QTP), Hu et al., (2019); Yang and Zhang (2018) reported that the root mean squared error (RMSE) of daily soil temperature from different reanalyses (i.e. ERA-Interim/Land, MERRA-2, and CFSR) ranged between 1.8-5.1 °C. This error is most often expressed as a cold bias."

P2, L40: ... and example numerical or process-based simulation... Revised.

P2, L57: Note that ERA5L is now available from 1981. In the revision, we added "Note that at the time of writing, only ERA5L data after 2001 have been released to the public and so this evaluation is conducted using data between 2001–2018.".

P4, L86: The soil temperature from the TTOP and CP maps are used as comparisons, please as mention here.

In the revision, we added "The mean annual ground temperatures (MAGT) from the TTOP and CP maps were also used to evaluate ERA5L."

P4, L89: ...(denoted as PZI map)",".., should it be ";"? Similar in L91. Revised.

P4, L97: The MAGT of TTOP and CP maps are additionally used as reference in your Table 1 and Figure 3. Please clarify here.

Mentioned, see the response to comment on P4, L86.

P5, L104: ...in the same ERA5L grid cell... Revised.

P5, L107: ...of ERA5L soil temperature.... Revised.

P5, L126: there is a repeat of the "the". Revised.

P5, L134: ...and (2) an increase of 1 m wSDmax Revised.

P7, L149: Is the ALT also overestimated in high latitudes and underestimated in high altitudes? It is difficult to say as most of the sites in mid-low are excluded before evaluation since their ALT > 1.89 m. In this case, the evaluation shown here are generally for high latitudes (see Figure 5 for the site distribution of ALT < 1.89 m). We changed the caption of Figure 4 to "*The observed sites are mainly located in high latitudes, and the distribution is present in Figure 5.*" to clarify.

P10, L164: Also mention the high spatial (and maybe temporal) resolution here, this is one of the most significant features of ERA5 compared to the others.

Revised as "ERA5L has a number of advantages for permafrost research; it provides a long historical record (back to 1950, eventually), high spatial resolution, and global coverage."

P13, L215: ...for  $c_{\xi}$  in Eq. B5... Revised.

P13, L216: It should be 150 kg m-3 based on Eq. B5, please double check. Revised to 150 kg m-3.

P14, L236: Underestimate permafrost...(what)? Permafrost area? Please clarify. It is permafrost area, and is clarified in the revision.

P14, L252–253: The bracket is incomplete Revised.

P14, L255: Brackets are needed here for the url. Revised.

P15, L270: Add space between m and  $s^{-2}$  Added.

P16, L278:  $\rho_{\xi}$  is not included in Eq. (B5). The sentence ischanged to "where the  $a_{\xi}$ ,  $b_{\xi}$ , and  $c_{\xi}$  are constant values of  $2.8 \times 10^{-6}$  (s-1), 0.042 (-) and 460 (m3 kg-1) derived or modified from Anderson(1976) and Jordan et al. (1999)."

P16, L280: Considering move  $\Delta\beta_s = 0$  to the upper so that Eq B6 would be aliened with the state of Eq. B8 and B10 Revised. P16, L297: ...ice density of 920... Revised.

**Specific comments:**

- Table 1: This is only for the observations in permafrost regions. Please clarify in the caption otherwise including the observations in non-permafrost regions. Yes, this is only permafrost regions. The caption is changed to "Summary of soil temperature observations in permafrost regions..."
- Figure 3: In the caption, it should be "...(observation-ERA5L)..." Revised.
- Figure 6: Considering add unit to the permafrost area changing rate. Note that the original Figure 6 is deleted, as the ERA5L simulated soil temperature, therefore permafrost area, is not well.

**Responses of Anonymous Referee #3**

This paper assesses the utility of ERA5L soil temperature products for permafrost studies by using a wide range of global station data from both permafrost and non permafrost regions as well as detailed simulation experiments at a specific site. The authors find that ERA5L has large biases making the product problematic for permafrost studies. This study is a valuable contribution as we increasingly use reanalysis products for land surface modeling studies, especially at regional or global scales and insights into performance of these products are useful. Additionally, such studies may help to guide future developments in land surface schemes used in reanalyses. I recommend publishing after considering my (mainly minor) comments. (in grammatical comments changes are CAPITALIZED)

The manuscript has been carefully edited by native speaker with strong permafrost background in order to improve the language.

1. 1.3 "is predicted TO BE too warm...." Revised as "We find that ERA5L overestimates soil temperature in northern Canada and Alaska...".

2. 1.19 "Reanalysis, ASSIMILATES" Revised as *Reanalysis consists of assimilating a broad range...*

9. 1.74 "These INCLUDE" Revised as "*Of these, there are...*".

13. 1129 "A linear model..." Revised as "*The following linear model*...".

16. l.147 "While ERA5L does not have DATA allowing deep ALT values to be computed" Revised as "While ERA5L is not capable of representing deep ALT...".

26, 1.232. use of "low" here is confusing. you are biased to low densities, you do not have a low bias. I would say "a low-density bias" to make it clear.

We would keep the sentences as its current format: "ERA5L snow density is hypothesized to have a low bias, at least in high-latitude areas, explaining part of the warm bias in soil temperature." as the snow do have a low bias.

3. 1.28 what is ERA5-Interim/Land? Seems a confusion of the products It was a typo, should be ERA-Interim/Land.

4. 1.29 "consistently cold BIASED." Revised to "*This error is most often expressed as a cold bias.*"

5. 1.54 I think the HTESSEL ref could do with a publication citation. The latest ERA5 paper, Hersbach et al., (2020), that describes HTESSEL is added here.

6. 1.57 now available from 1981.

In the revision, we added "Note that at the time of writing, only ERA5L data after 2001 have been released

to the public and so this evaluation is conducted using data between 2001–2018.".

7. Section 2.2.1 what do B1 and B2 refer to? Revised to "*Appendix B1*" and "*Appendix B2*" to clarify.

8. 1.71 is the node really at the lower boundary (0.07) in soil layer 1? Reviewer #2 also mentioned this issue. We copied the responses here.

**Based on the Table 8.7 in IFS Documentation CY45R1**

(https://www.ecmwf.int/en/elibrary/18714-part-iv-physical-processes), the lower boundary is 0.07 m, although this is different from the description in ERA5L document website). We also contacted the scientist from ECMWF, and I simply copied the reply below.

"The soil temperature of a given layer is an averaged value over of the thickness of that layer and assigned to the middle of the layer. From the modeling point of view this temperature is a valid temperature for any point in the layer, whereas in reality it'll be different depending on the depth. This is one of the limitations when the soil is discretised in a finite number of layers."

For this reason, we followed the depth in model document as described in L70–71: "*The soil column of ERA5L is discretized into four layers with node depths (layer boundaries) of* 0.07 (0–0.07), 0.21 (0.07–0.28), 0.72 (0.28–1.00), and 1.89 (1.00–2.89) m."

10. Section 2.3 and Table 1 are all stations boreholes? If so perhaps explicitly state that. No, some sites are from boreholes, e.g., GTN-P, but sites like CMA, HiWATER, are from soil temperature sensor of meteorological stations. In Section 2.3, we added "*Sites consist of both meteorological stations and boreholes*." to clarify..

11. 1.90-91 and driven by ERA-Interim air temperature.

The TTOP map compiled by Obu et al., (2019) was driven mainly by MODIS LST, but the data gaps due to cloud cover was filled by downscaled ERA-Interim air temperature. Please see Section 2.2 and Figure 1 from Obu et al., (2019). In the revision, we changed the sentence to

"...(3) the 1-km Northern Hemisphere permafrost map Obu et al., (2019) which is based on the semi-physical Temperature at the Top Of Permafrost (TTOP) model (TTOP map) driven by Moderate Resolution Imaging Spectroradiometer (MODIS) land surface temperature that filled by downscaled ERA-Interim air temperature;" to clarify.

12. 1111-114: I don't quite understand the motivation for the two definitions of near surface permafrost I think a sentence explaining why you do this would be helpful for the reader.

The two algorithms are defined here to derived ERA5L soil temperature-based permafrost area. In the revision, this sentence is changed to "The ERA5L near-surface permafrost area is evaluated using existing permafrost maps."

14. 1.137 What depth are these MAGT's? Averaged across time or space? Please provide a bit more detail here.

Reviewer #1 also had similar comment. We copied the response here as well.

In Section 3.1, we added "The TTOP and CP map were derived using an equilibrium model, and MAGT is given as an average of the entire period (MAGTavg). This corresponds to 2002–2014 for the CP map and 2002–2016 for the TTOP map, without uniform/specific soil depth. To better evaluate, we aggregated all available observed MAGTs during the period by averaging, and then compared against the MAGTavg of these two maps. Note that the performance of CP and TTOP maps may be lower here than reported in the original publications due to the fact that we evaluate them with a different set of observations (different depths, periods and proportion of sites in mountains." to clarify

15. 1.143 more prevalent snow and soil freezing in the model or in reality? Please clarify. If in reality, then permafrost regions do not necessarily have more prevalent snow than non-permafrost regions.

This is in the HTESSEL model or ERA5L based on the bias comparison of in permafrost and non-permafrost regions(Table 2; Figure 1). We revised this part to "In addition to the worse performance of MAAT in these regions, the result suggests that HTESSEL may be less suitable for soil temperature simulation in areas with more prevalent snow and soil freezing. The large warm bias of ERA5L soil temperature during winter (Fig-

Figure 2: Comparison of shallow active layer thickness (ALT) based on 787 measurement from 106 stations located in 79 grids. The observed sites are mainly located in high latitudes, and the distribution is present in Figure 5.

ure 2) further supports this notion."

17. 1.153 "(annually)" is it an annual average? Please clarify.

Yes, it is revised to "Near-surface permafrost area of ERA5L as defined in this study decreased at a rate of  $-0.11 (-0.08) \times 10^6 \text{ km}^2 \text{ year}^{-1}$  based on hourly (annual) mean soil temperature. This corresponds to a loss of  $1.7 (1.4) \times 10^6 \text{ km}^2$  of near-surface permafrost area since 2002. This is also suggested by the linear model." to clarify.

18 Figure 3 Interesting latitudinal trend in c,d. Can you shed more light on this in the discussion? I guess densification processes at high latitudes (badly represented wind?) What is driving the cold bias at low latitudes?

As Reviewer #1 mentioned, HTESSEL does not have a representation of wind effects on snow densification. In this case, blowing snow. Both Figure 1 and the linear model (Eq. 1) indicated that the cold bias at low latitudes is largely due to the MAAT bias. In Section 5.1, we added

"Our results indicate that the cold bias of ERA5L in mid-low latitudes is highly aligned with the MAAT bias. This is also suggested by the linear model (Eq. 2)."

19. Figure 4 perhaps add the mean value that you cite in the text here. Revised (see Figure 2 here).

20. 1.170 "shows REMARKABLY" Revised as "ERA5L shows a remarkable underestimation of total permafrost area".

21. 1.194 "Even for A" Revised.

22. 1.198 "This issue is KNOWN" Revised.

23. 1.208 "as AN exponential..." Revised.

24. 1.226 soil temperatureS MATCH..." Revised as suggested.

25. 1.230 But what about the cold bias you see? the bias appears to evenly spread (figure3) why does this not give a similar spread in ALT estimates (Figure 4) and a related underestimation of ALT?

This is because the shallow ALT sites (< 1.89 m) are mainly in high latitudes (Figrue 5), and in high latitudes the soil temperature was found too warm. This is aligned with Figure 3. In the revision, we changed the caption of Figure 4 to "*
[revised manuscript text omitted]

| Source      | Ν   | Coverage           | SL (depth)                  | Reference               |
|-------------|-----|--------------------|-----------------------------|-------------------------|
| СМА         | 56  | 2001–2006 (-26–38) | 1-4 (0.05-1.60)             | Wang et al. 2015        |
| WDC         | 105 | 2001-2015 (-40-30) | 2-4 (0.02-1.60)             | _                       |
| Nordicana D | 219 | 2001–2018 (-42–25) | 1-4 (0.00-2.10)             | _                       |
| GI-UAF      | 95  | 2001-2018 (-40-23) | 1-4 (0.01-2.00)             | Wang et al. 2018        |
| Tibet-OBS   | 10  | 2008-2016 (-18-28) | 1-3 (0.05-0.40)             | Su et al. 2011          |
| CTP-SMTMN   | 60  | 2010-2016 (-15-20) | 1-3 (0.04-0.40)             | Yang et al. 2013        |
| GTN-P       | 40  | 2001-2018 (-41-26) | 1-4 (0.00-2.40)             | Biskaborn et al. 2015   |
| NPS         | 28  | 2004–2016 (-33–24) | 2-3 (0.20-0.75)             | Wang et al. 2018        |
| USGS        | 16  | 2001-2015 (-31-25) | 1-2 (0.05-0.20)             | Urban and Clow2017      |
| HiWATER     | 8   | 2012–2017 (-19–22) | 1-3 -1-4 (0.04-2.00) | Che et al. 2019         |
| Others      | 2   | 2001-2018 (-32-14) | 1-4 (0.04-1.95)             | Boike et al. 2018; 2019 |

downscaled ERA-Interim air temperature; and (4) the 1-km circumpolar permafrost map (CP map) which is derived from a statistical model (Karjalainen et al., 2019a). While-

Whereas ERA5L, TTOP and CP maps represent permafrost distribution with binary information as a boolean variable (i.e.
presence or absence based on present or absent according to soil temperature), the IPA map and PZI map use categories represent permafrost using either a categorical variable (e.g., continuous, discontinuous, sporadic, and or isolated permafrost) or a continuous index (0.01–1) to as a proxy to approximately represent the proportion of an area underlain by permafrost (i.e. the permafrost extent). By following Melton et al. (2019), Following Melton et al. (2019), we apply a threshold of 50% (corresponding to the continuous and discontinuous permafrost zones) and 0.5 for the IPA map and 0.5 for the PZI map, respectively, are used for permafrost area estimation and for comparing areaswith binary maps . to allow for meaningful comparison with the other maps. Values greater than this are considered to represent permafrost areas. The mean annual ground temperatures (MAGT) from the TTOP and CP maps were also used to evaluate ERA5L.

**3 Method**

**3.1 Evaluation**

130 The observed temperatures are grouped by depth according to the four For the purposes of evaluation, temperature observations were only used from depths between 0 m and 2.89 m, corresponding to the range of the ERA5L soil column. Temperature values were grouped according to their depth into one of the ERA5L soil layers. For the layer with observations from multiple depths

When this mapping resulted in multiple depths being assigned to a single soil layer, the one nearest to the ERA5L grid center is selected. was selected. The ERA5L soil temperature is temperatures were nearest-neighbour interpolated to observed each

- 135 of the observation sites to avoid the missing values of adjacent water bodymissing values caused by adjacent water bodies. The mean bias (BIAS), mean absolute error (MAE), and RMSE were used for comparison against observations at as metrics to compare observations to ERA5L at the station scale (see appendix A). As multiple sites could be In the case where multiple sites were located in the same ERA5L grid cell, BIAS, MAE, and RMSE are were calculated for each site individually and then aggregated for each unique grid by averaging with equal weights by averaging all stations in each grid cell . In this context,
- 140 weighted metrics, for example wBIAS, is used for with equal weight. For the evaluation at ERA5L grid scale., these aggregate metrics (for example, wBIAS) were used.

MAAT bias and maximum snow depth  $(SD_{max})$  were selected as candidate variables to be assessed as possible predictors of ERA5L temperature bias.  $SD_{max}$  is derived soil temperature bias (see Eq. 1).  $SD_{max}$  was defined as the median of annual maximum monthly snow depth during the period 2001–2018. Surface The surface offset (SO), which quantifies the influence

145 of surface conditions <del>, e.g., such as snow and vegetation cover</del> (Smith and Riseborough, 2002), and is derived is defined here as the difference of MAAT and mean annual ground temperature (MAGT ) of soil layer 1 between MAAT and MAGT of the uppermost soil layer in ERA5L.

ERA5L ALT was derived by linearly interpolating the ERA5L soil temperature-depth profiles. The TTOP and CP map were derived using an equilibrium model, and MAGT is given as an average of the entire period ( $MAGT_{avg}$ ). This corresponds to

- 150 2002–2014 for the CP map and 2002–2016